# Unsupervised pipeline for the identification of cortical excitatory and inhibitory neurons in high-density multielectrode arrays with ground-truth validation

Eloise Giraud[1], Michael Lynn[2,3], Philippe Vincent-Lamarre[1,2], Jean-Claude Beique[2,3], Jean-Philippe Thivierge[1,2]*

[1]School of Psychology, University of Ottawa, Ottawa, Canada; [2]uOttawa Brain and Mind Research Institute Center for Neural Dynamics, University of Ottawa, Ottawa, Canada; [3]Department of Cellular and Molecular Medicine, Faculty of Medicine, University of Ottawa, Ottawa, Canada

**\*For correspondence:**
jthivier@uottawa.ca

**Competing interest:** The authors declare that no competing interests exist.

## eLife Assessment

In this manuscript, the authors describe a software package for automatic differentiation of action potentials generated by excitatory and inhibitory neurons, acquired using high-density microelectrode arrays. The work is **valuable** as it offers a tool with the potential to automatically identify these neuron types in vitro. It is **solid**, as it provides a tool to identify putative excitatory and inhibitory neurons on high-density electrode arrays, which can be used in conjunction with other existing spike sorting pipelines.

**Abstract** Large-scale extracellular recording techniques have advanced the study of neuronal circuits but lack methods to reliably identify cell types while scaling to thousands of neurons. We introduce spikeMAP, a pipeline for analyzing large-scale in vitro cortical recordings that combines spike sorting with cell-type identification using viral and optogenetic validation. SpikeMAP integrates data analysis with optogenetic, viral, and pharmacological protocols to dynamically probe distinct cell types while recording from large populations. The pipeline fits spike waveforms using spline interpolation to measure half-amplitude and peak-to-peak durations, applies principal component analysis and k-means clustering to isolate single-neuron signals, and uses linear discriminant analysis to optimize cluster separability. Channel source locations are determined through spatio-temporal spike waveform characteristics. Applied to mouse prefrontal cortex slices recorded on a 4096-channel array, spikeMAP effectively distinguishes regular-spiking excitatory neurons from fast-spiking inhibitory interneurons via action potential waveform, Fano factor, and spatial cross-correlations. This validated toolbox enables comprehensive characterization of neuronal activity across cell types in high-density recordings, offering a scalable approach to study microcircuit interactions in the brain.

## Introduction

The advent of high-density, large-scale multielectrode array recordings has advanced our understanding of how populations of neurons represent and process information, offering crucial insights into complex network dynamics (*Riquelme et al., 2023*; *Hemberger et al., 2019*; *Shein-Idelson et al., 2017*; *Thivierge et al., 2022*). Furthermore, the use of silicon recording probes such as Neuropixels probes in vivo has helped uncover foundational principles of neural circuits in awake behaving mice (*Lak et al., 2020*; *Peters et al., 2021*). However, it remains a challenge to distinguish neuronal cell types based on their physiological properties alone, despite a growing appreciation for the importance of heterogeneous cell types in circuit-level computations (*Sylwestrak et al., 2022*; *Liu et al., 2021*; *Bugeon et al., 2022*). The scalability of large-scale electrophysiological recording techniques depends not only on the detection and isolation of spikes produced by single neurons (spike sorting) but also on the reliable distinction between those triggered by different cell types.

The recording of extracellular potentials is a popular method to investigate neural activity, offering a high spatiotemporal resolution while allowing for the monitoring of action potentials from thousands of cells simultaneously (*Magland et al., 2020*; *Buzsáki, 2004*). Cell-type sorting based on extracellular action potential waveforms has been extensively explored. It has been proposed that excitatory (E) and inhibitory (I) neurons can be distinguished based on action potential waveforms (*Hafizi et al., 2022*; *Ren et al., 2020*; *Barthó et al., 2004*). These methods typically extract features of action potentials such as peak-to-peak time difference and full-width-half amplitude maximum and match these features to reported values for each cell type to distinguish them. The rationale for using these features as a proxy for cell-type identity stems from intracellular recording studies, which report that inhibitory cells, such as parvalbumin-expressing GABAergic interneurons, have short-duration action potentials (*Nowak et al., 2003*; *Peyrache and Destexhe, 2019*; *Kawaguchi and Kubota, 1993*; *Connors and Gutnick, 1990*), whereas excitatory pyramidal neurons have longer-duration action potentials (*McCormick et al., 1985*). Narrow and broad action potentials have been distinguished from one another in several neocortical areas, including prefrontal cortex (*Mitchell et al., 2007*; *Wilson et al., 1994*; *Constantinidis and Goldman-Rakic, 2002*; *González-Burgos et al., 2005*; *Hasenstaub et al., 2005*; *Barthó et al., 2004*; *Tamura et al., 2004*). These differences have been hypothesized to result from the fact that repolarization of the membrane potential following a spike is slower among broad-spiking pyramidal neurons (*McCormick et al., 1985*; *Nowak et al., 2003*; *Hasenstaub et al., 2005*). In the extracellular signal, which resembles the negative derivative of the

**Table 1.** **lang**.: Programming language, **# elect**.
: Maximum number of electrodes tested on, **data preproc**.: Data preprocessing (formatting, filtering, artefact removal) included, **event/channel loc**.: Localization of single events, **waveform extrac**.: Spike features (such as peak-to-peak distance and full-width half max) are extracted from waveforms, **e/i sorting**: excitatory and inhibitory cell sorting, **validation**: validation of cell-type sorting (*+*)=*ground truth validation via intracellular recordings, but not e/i validation*, **experim. prot**.: integrates an experimental protocol to allow for network manipulation, **unsuper**.: unsupervised, requiring no manual user intervention, **source code available**: entire code is open-source and available.

| | lang. | # elect. | data preproc. | event/ channel loc. | waveform extrac. | e/i sorting | validation | experim. prot. | unsuper. | source code available |
|---|---|---|---|---|---|---|---|---|---|---|
| *spikeMAP* | Python and Matlab | 4096 | + | + | + | + | + | + | + | + |
| Kilosort4 (*Pachitariu et al., 2023*) | Python | | + | + | - | - | - | - | - | + |
| SpyKING CIRCUS (*Yger et al., 2018*; *Yger et al., 2023*) | Python | 4225 | - | + | + | - | (+) | - | + | + |
| MountainSort (*Chung et al., 2017*) | Matlab and C++ | 16 | + | + | - | - | - | - | + | + |
| Herding Spikes (*Hilgen et al., 2017*) | Python | 4096 | + | + | + | - | (+) | + | + | + |
| YASS (*Lee et al., 2017a*; *Lee et al., 2017b*) | Python | 512 | - | + | - | - | - | - | + | + |
| JRCLUST (*Jun, 2017*, *Jun, 2017*) | Matlab and CUDA | 120 (1000) | + | + | - | - | (+) | - | - | + |

intracellular membrane potential, this gives both a broader and shallower peak following the initial trough (*Henze et al., 2000*).

Modern extracellular recording techniques allow researchers to monitor the simultaneous activity of large, intact networks of neurons. However, in dense preparations, the activity of a single cell may be captured by several electrodes, and each electrode may report the activity of several nearby cells. Thus, to segregate the activity of single cells, effective spike-sorting techniques must be employed. Spike sorting typically includes filtering the raw extracellular activity, detecting action potentials using threshold crossings, extracting spike features, and clustering to distinguish amongst neurons (*Rey et al., 2015*).

Several methods have been developed, each with specialized features, including Kilosort4 (*Pachitariu et al., 2023*), SpyKING CIRCUS (*Yger et al., 2018*), MountainSort (*Chung et al., 2017*), Herding Spikes (*Hilgen et al., 2017*), YASS (*Lee et al., 2017a*), and JRCLUST (*Jun et al., 2017*). A detailed comparison of these methods is presented in *Table 1*. A common lacking feature in these methods is the ability to identify cell types in a biologically validated fashion. Despite evidence showing differences in action potential kinetics for distinct cell types as well as the use of optogenetics (*Hilgen et al., 2017*), there exist no large-scale validation efforts, to our knowledge, showing that extracellular waveforms can be used to reliably distinguish cell types. Methods leveraging genetic differences between cell types offer statistical measures predictive of cell types, such as Fano Factor (*Becchetti et al., 2012*). These methods have provided researchers with the ability to infer, in a post-hoc fashion, the most probable cell types that were sampled. Yet, a fully integrated suite offering an open-source data analysis pipeline paired with a validated experimental protocol allowing for high-density cell-type-specific activation during recordings is needed. Such a method will enable the investigation of dense and intricate circuits in a flexible, scalable, and generalizable manner.

Here, we introduce spikeMAP, an open-source, unsupervised and scalable spike sorting analysis pipeline which performs cell-type classification as validated on high-density MEAs. Crucially, we tightly integrate our analysis with an experimental protocol utilizing a stabilized step function opsin to dynamically probe distinct populations and validate sorting while avoiding the light-induced artefacts common on CMOS-based chip designs (*Fiscella et al., 2012*). Our analysis pipeline, taking advantage of dense sampling, estimates cell-body location for each detected spike, yielding well-separated clusters associated with single neurons. Spike waveform features were extracted utilizing signal interpolation to estimate half-amplitude and peak-to-peak durations. These values were then entered in a principal component analysis with k-means clustering to identify uncorrelated signals from single channels on the array. Optimal separability of clusters was assessed by linear discriminant analysis. Each channel's source location was identified using spatiotemporal characteristics of spike waveforms across the array. We further classified cells as putative excitatory or inhibitory based on their waveform properties. As a final validation, we introduced a method taking advantage of long timescale optogenetics (step function opsins) to perform cell-type-specific stimulation in acute prefrontal cortex slices from mice, providing a biological verification to our statistical methods. Together, spikeMAP provides

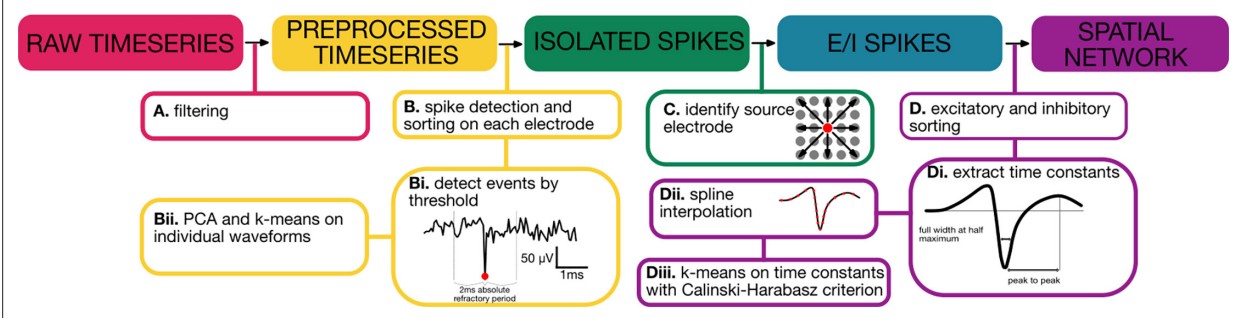

**Figure 1.** Workflow diagram of the proposed method. (**A**) Filtering of the data is achieved through bandpass filtering (300–5000 Hz). Spike detection is done on filtered voltages; however, raw voltages around each spike are used to avoid signal distortion. (**B**) Spike sorting on each electrode. (**Bi**) Events are detected using a threshold method (6 SD from the mean of the filtered signal). (**Bii**) PCA and k-means on individual waveforms (k=2). (**C**) Precise soma location on the array. (**D**) Excitatory and inhibitory sorting. (**Di**) Spike time constant kinetics are extracted at the full width at half maximum and peak to peak distances. (**Dii**) Interpolated voltage for increased resolution. (**Diii**) k-means to distinguish cell type based on their waveform kinetics.

a comprehensive toolkit for experimenters to flexibly interrogate cell-type-specific computations in a variety of neural circuits throughout the brain.

## Results

### Spike sorting

Our spike sorting pipeline (*Figure 1*) performs three main tasks: (i) the detection of voltage deflections associated with putative somatic spikes, (ii) the identification of individual putative neurons using spike clustering, and (iii) the spatial localization of single cells in high-density recordings. We demonstrate the use of this pipeline through targeted high-density recording from medial prefrontal cortex of mouse (*Figure 2A*; see Materials and methods). First, putative spike times were detected as threshold crossings (*Figure 2Bi*; *Lewicki, 1998*), identifying negative signal peaks from the mean of the filtered signal (see Materials and methods). Optimal threshold choice is key in discriminating spikes from noise, as the number of spikes detected per electrode is highly dependent on the threshold used (*Figure 2C*). Here, the optimal threshold for detection was selected by fitting the voltage histogram of each channel with a Gaussian distribution to separate spiking activity from normally distributed noise (*Figure 2D*). At a threshold of –3σ, the signal could be largely accounted for by Gaussian noise, while a separation between signal and noise began around a threshold of –4σ. To avoid multiple detections of the same event, an absolute refractory period was imposed to discard successive events within <2ms. Although filtering is crucial in event detection, it can introduce signal distortion (*Yael and Bar-Gad, 2017*). To resolve this issue, the unfiltered voltage around the time of each spike was extracted (*Figure 2Bii*), yielding a series of high-frequency (18 kHz) waveforms for each channel.

In high-density preparations, a single electrode can report the activity of multiple cells. To address this issue, we employed a principal component analysis on the identified spike waveforms from individual channels (*Rossant et al., 2016*), which revealed distinct clusters putatively corresponding to individual cells (*Figure 2E*), indicating that more than one neuron likely contributes to the voltage recorded on some channels. The SpikeMAP suite also offers a routine to select a radius around individual channels in order to enter groups of adjacent channels in PCA. Next, we employed a k-means cluster analysis to extract clusters amongst the principal components. To determine how well the data was separated by this approach, we applied a linear discriminant analysis (LDA) to the labeled principal components (*Hill et al., 2011*), allowing our method to remain completely unsupervised. In SpikeMAP, the optimal number of k-means clusters can be chosen by a Calinski-Harabasz criterion (*Calinski and Harabasz, 1974*) or pre-selected by the user.

To ensure the activity of each single cell was only reported once, we set a radius of 250 µm around the electrode where a given cell was detected. This radius corresponds to the approximate spread of extracellular electrical field around the soma of individual neurons (*Jia et al., 2019*). Within this radius, the channel with the largest negative deflection in mean waveform was identified. Previous work observed that the largest negative amplitude of extracellular signal originates from the axon initial segment (AIS; *Bakkum et al., 2019*; *Obien et al., 2014*). Utilizing spatial localization of the AIS, the channel showing the greatest negative deflection was likely the closest to the putative soma of a given cell (*Jia et al., 2019*). Activity at this channel was consistent with the somatic activity, showing a decay in peak voltage as we moved away from the putative soma (*Figure 2H*; *Delgado Ruz and Schultz, 2014*; *Pettersen and Einevoll, 2008*; *Lindén et al., 2011*; *Mechler et al., 2011*; *Figure 3*). Visualization of adjacent electrodes revealed a complex structure of voltage activity, possibly due to activity propagation within the compartments of single neuron (*Figure 2F*; *Shein-Idelson et al., 2017*; *Obien et al., 2014*). As an additional verification step, SpikeMAP allows the computation of spike-count correlations between putative neurons located within a user-defined radius. Signals that exceed a defined threshold of correlation can be rejected as they likely reflect the same underlying cell. To further verify that single cells located near the same electrode could be distinguished, we estimated the precise soma locations between adjacent recording electrodes using a center-of-mass analysis (see Materials and methods). This revealed subtle changes in position, reflected by distinct weights on each nearby electrode (*Figure 2I*). Despite these subtle changes, the soma location could still be assigned to the same electrode since changes in localization across spikes were far less than the inter-electrode distance.

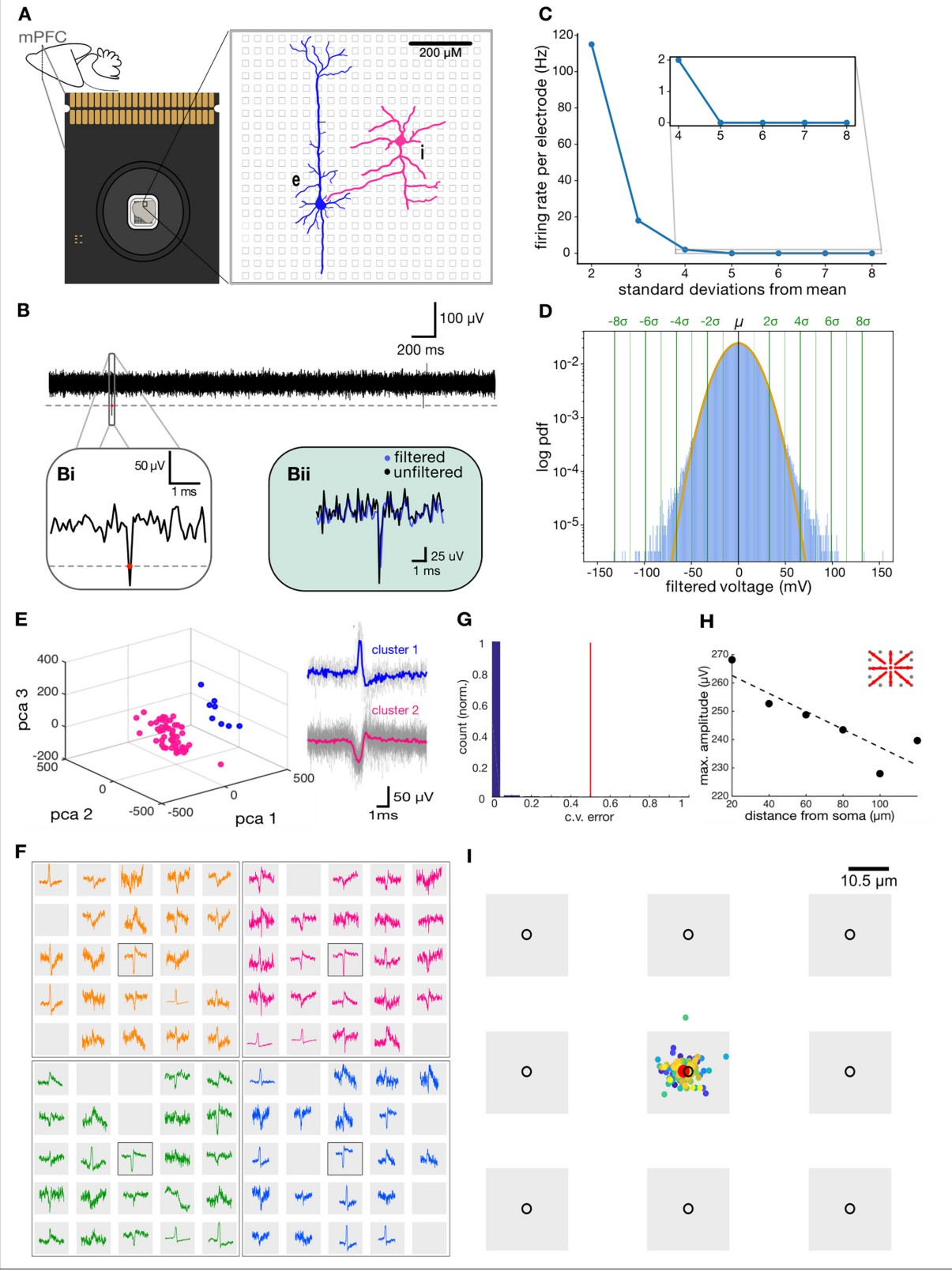

**Figure 2.** Isolation of single-cell action potential for each electrode. (**A**) Multielectrode array recording set up. (**B**) Extracellular voltage on a single electrode. (**Bi**) Spike detection on filtered voltage. (**Bii**) Raw voltage extraction around detected spike. (**C**) Firing rate per electrode depends on threshold selection. The inset shows firing rate values at the selected threshold (4 SD). (**D**) Threshold selection to part cellular activity from normally distributed noise. (**E**) Spike sorting on a single electrode. (**F**) Current sink signal from single neuron spans multiple electrodes (four different neurons are

*Figure 2 continued on next page*

*Figure 2 continued*

shown in different colors), (**G**) Histogram distribution of tenfold cross-validation with chance performance indicated in red for all electrodes in a single recording. (**H**) Voltage amplitude as a function of distance from highest electrode peak, inset shows the directions measured from the center electrode. (**I**) Center of mass location of all spikes for a single soma, mean is indicated in red.

In sum, SpikeMAP provides an end-to-end pipeline to perform spike-sorting on high-density multielectrode arrays. Some elements of this pipeline are similar to related approaches (*Table 1*), including the use of voltage filtering, PCA, and k-means clustering. Other elements are novel, including the use of spline interpolation, LDA, and the ability to identify putative excitatory and inhibitory cells. At different steps in the process, conditions for rejecting spikes can be tailored by applying: (1) a stringent threshold to filtered voltages; (2) a minimal cut-off on the signal-to-noise ratio of voltages (*Figure 4*); (3) an LDA for cluster separability; (4) a minimal spike rate to putative neurons; (5) a Hartigan statistical dip test to detect spike bursting; (6) a decrease in voltage away from putative somas; and (7) a maximum spike-count correlation for nearby channels. Together, these criteria allow SpikeMAP users the ability to precisely control parameters relevant to automated spike sorting.

## Cell type classification based on action potential waveform kinetics

Despite technological advances, the ability to identify distinct cell types based solely on extracellular voltage has remained challenging. Current data processing methods often rely on waveform kinetics (*Barthó et al., 2004*) to return a classification of contributing excitatory and inhibitory units. As technological progress continuously allows for denser arrays, waveform spike analyses are increasingly sensitive to noise generated by the signal of proximal units. Here, we first implemented a classification method using current knowledge about cell-type-specific waveform kinetics and spline interpolation to distinguish putative excitatory and inhibitory neurons in dense networks. We then validated this classification using a combination of viral strategies, optogenetics, and pharmacology while simultaneously recording from stimulated mouse prefrontal cortex networks.

To provide a more precise description of the shape of each spike, we first fitted each waveform using piecewise cubic spline interpolation, a method allowing for the statistical inference of points within the boundaries of a set of known points. While several interpolation bases are possible, cubic splines allow for smooth interpolation between known data points and a high accuracy in terms of curve fitting (*Ramsay and Silverman, 1997*). Using query points at 90 kHz, we computed the mean fitted waveform (*Blanche and Swindale, 2006*) for each putative cell (see Materials and methods). While we found that a resolution of 90 kHZ provided a reasonable estimate of spike waveforms, this value can be adjusted as a parameter in SpikeMAP.

Next, we identified putative excitatory and inhibitory neurons using two time constants obtained from mean spike waveforms at somatic locations: (1) the full width (FW) of spikes at half-maximum amplitude and (2) the peak-to-peak (PP) duration (*Figure 5A*). (*Sirota et al., 2008*; *Sakata and Harris, 2009*; *Barthó et al., 2004*; *Insel and Barnes, 2015*). Although some work has cast doubt on the accuracy of spike-width-based classification (*Moore and Wehr, 2013*), many studies have shown good classification accuracy (*Nowak et al., 2003*; *Kawaguchi and Kubota, 1997*; *Barthó et al., 2004*;

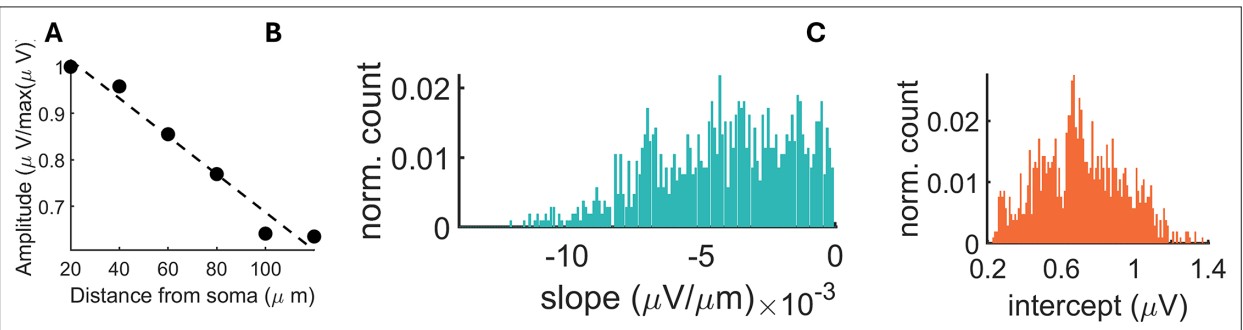

**Figure 3.** Decrease in voltage relative to the distance from putative soma. (**A**) Voltage amplitude as a function of distance from highest electrode peak. Putative somas with an increase in voltage away from electrode peak were excluded from this analysis. (**B–C**) Distribution of slopes and intercepts across all putative somas obtained from one recording (N=1950).

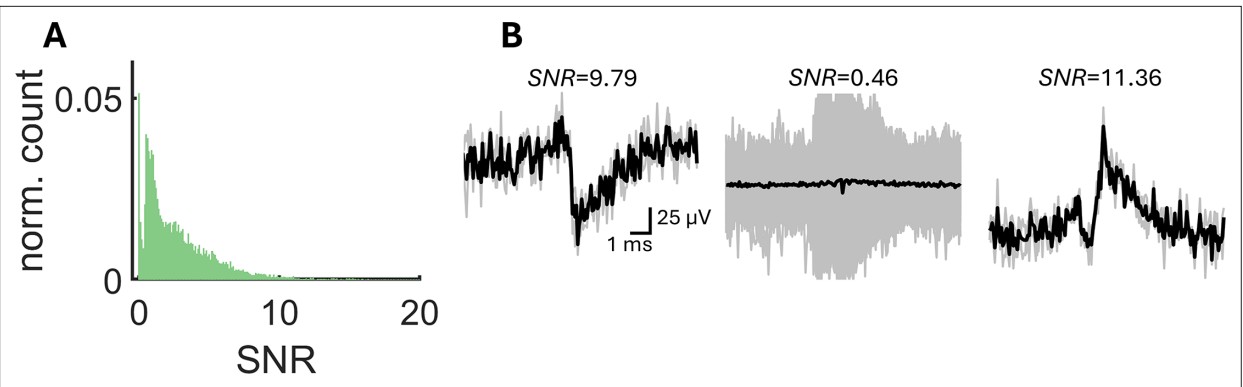

**Figure 4.** Analysis of signal-to-noise ratio for single channels. (**A**) Distribution of signal-to-noise ratio (SNR) over all spike-sorted clusters of a single recording (N=8059). (**B**) Examples of mean voltages (black lines) and individual voltages (gray lines) around individual spikes.

*Madisen et al., 2012*; *Stark et al., 2013*; *Cardin et al., 2009*; *Cohen and Mizrahi, 2015*). K-means clustering was performed on values of FW and PP obtained across all simultaneously recorded somatic electrodes, where the optimal number of clusters was chosen with a Calinski-Harabasz criterion (*Calinski and Harabasz, 1974*; *Vendramin et al., 2009*). For validation purposes, strict criteria were applied at each step of the pipeline, resulting in the low number of neurons reported here. Clusters corresponding to putative excitatory (E) and inhibitory (I) neurons were identified based on FW and PP values (*Figure 5A*). I cells were identified as a single cluster with rapid time constants for both FW and PP (*Barthó et al., 2004*). We extracted the putative soma locations of excitatory and inhibitory neurons on the MEA (*Figure 5B*) and visualized firing activity in a raster plot (*Figure 5C*) (see *Figure 6* for firing rate distributions). It is possible that the spatial location of putative I cells reflects the site of injection of the opsin in medial prefrontal cortex. Other properties of putative I cells were consistent with past work, including a higher Fano factor (*Figure 5D*; *Becchetti et al., 2012*) and a rapid spatial decay of pairwise correlation (*Figure 5E*; *Peyrache et al., 2012*). Further, SpikeMAP contains a routine to perform a Hartigan statistical dip test on the inter-spike distribution of individual cells to detect putative bursting neurons. These results extend previous findings to large-scale recordings, yet a ground-truth validation of cell type is still required to assess the extent to which this approach offers a reasonable proxy of biological identity, as examined next.

## A step-function opsin validation approach for cell classification

While multielectrode arrays have allowed the simultaneous recording from thousands of electrodes, the reliance on light-sensitive complementary metal-oxide-semiconductor (CMOS) circuits has limited the ability of researchers to use them in conjunction with optogenetics. As the ability to rapidly perturb distinct populations in large networks provides valuable insight as to how local circuits represent information (*Lemon et al., 2021*; *Buzsáki, 2010*), it is of broad interest to validate methodologies that allow for optogenetic manipulation of defined cell populations in multielectrode arrays.

We validated an experimental protocol using a stabilized step-function opsin (Bi-stable neural state switches; *Berndt et al., 2009*), which is activated in a step-like fashion for prolonged periods (minutes) by brief photostimulation with blue light and inactivated by green-yellow light. This viral and optogenetic strategy, used in conjunction with CMOS-based multielectrode arrays, allows for the monitoring of induced spiking beyond the initial light-generated artefacts (*Figure 7*). We demonstrated this by inducing spiking selectively in targeted PV inhibitory neurons, as more than 90% of them have been reported to be fast-spiking (*Stringer et al., 2016*), to verify whether cells previously identified as putative inhibitory interneurons indeed correspond to light-activated cells.

As an obligate validation step, we injected a viral vector expressing the step function opsin, pAAV-Ef1a-DIO hChR2 (C128S/D156A)-EYFP, in medial prefrontal cortex of mice, and after waiting 2–3 weeks for transfection, performed whole-cell recordings from pyramidal neurons located near identified PV interneurons using in vitro slice preparations (*Figure 8A*; see Materials and methods). Photostimulation with blue light was sufficient to activate PV interneurons, as monitored by the robust, rapid, and consistent appearance of light-induced inhibitory synaptic postsynaptic currents

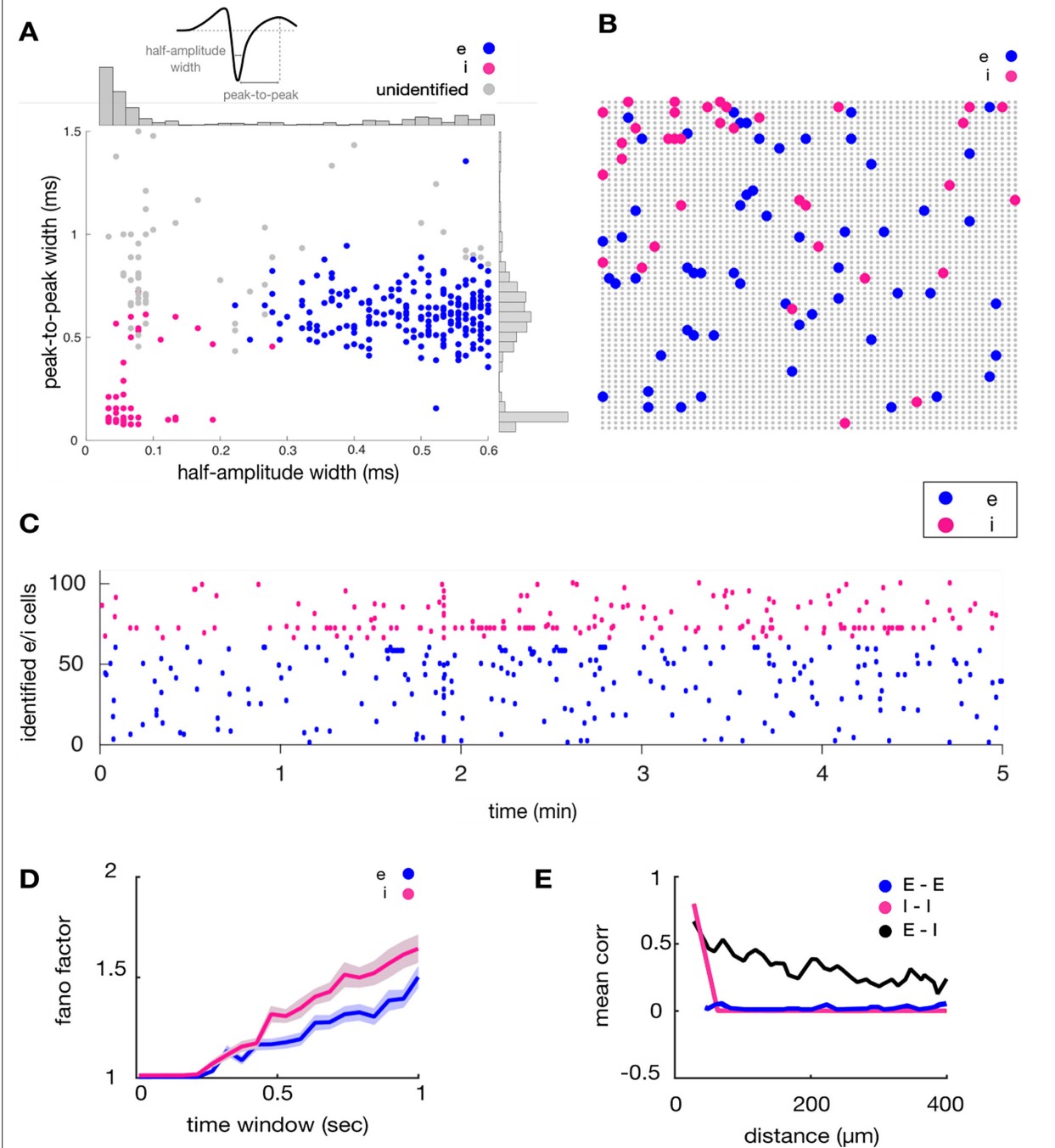

**Figure 5.** Identification of putative excitatory and inhibitory units. (**A**) Distinct waveform kinetics for E and I cells (**B**) Respective cell location (**C**) Raster plot of E and I activity. (**D**) Fano factor for putative E and I cells. The shaded area represents the SEM. The x-axis shows temporal bins of increasing durations employed to compute the Fano factor (**E**) Mean cross-correlation as a function of spatial distance between pairs of electrodes. Non-overlapping bins of 100ms were employed to compute correlations. The resulting values were averaged by spatial distance between all pairs of neurons.

(IPSCs) recorded from pyramidal neurons (note the absence of direct photocurrent from these cells; *Figure 8B*). The subsequent delivery of short pulses of yellow light reduced the frequency of IPSCs onto recorded pyramidal neurons, indicating that the step-function opsin could be reliably activated and inactivated in a step-like manner by phasic delivery of light (*Figure 8B*). Upon PV photoactivation, recorded pyramidal neurons reliably received high-frequency barrages of 10–30 pA IPSCs, followed by a gradual reduction in amplitude and elevated rates of inhibitory input that persisted until deactivation with green-yellow light, reflecting some desensitization of the step-function opsin (*Figure 8B*).

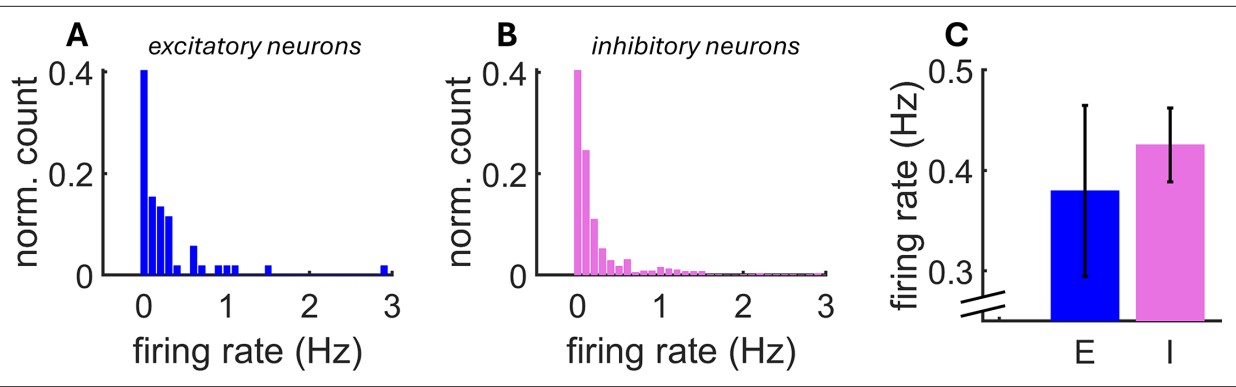

**Figure 6.** Distribution of mean firing rates across a population of putative excitatory (**A**) (N=570) and inhibitory (**B**) (N=54) neurons. (**C**) Mean firing rate across neurons. Vertical bars: SEM.

To characterize the refractory period of the step-function opsin, we delivered repeated photostimulation at various inter-trial intervals and measured steady-state activation (**Figure 8C** to E). Longer inter-trial intervals (≥120 s; **Figure 8E**) allowed expressing cells to recover and prevented the reduction in steady-state activation observed with 20 s and 60 s inter-trial intervals (**Figure 8C and D**). Specifically, the mean IPSCs/s for 20 s, 60 s, and 120 s intervals were 12.64±0.27, 17.32±0.15, and 26.29±0.31, respectively. Statistical comparisons, performed using the Wilcoxon rank sum test for non-parametric data on the PSTH, revealed significant differences between the 20 s and 120 s conditions (p=1.29 × 10$^{-6}$, n=4500 and 3000 trials, respectively) as well as between the 60 s and 120 s conditions (p=0.032, n=3000 trials per condition). The number of trials conducted for each condition was 5 for 20 s, 6 for 60 s, and 6 for 120 s.

Furthermore, the reduction in firing rate upon opsin deactivation was more prominent for longer interval trials (**Figure 8E**). These results provided key insights to inform multielectrode array recording

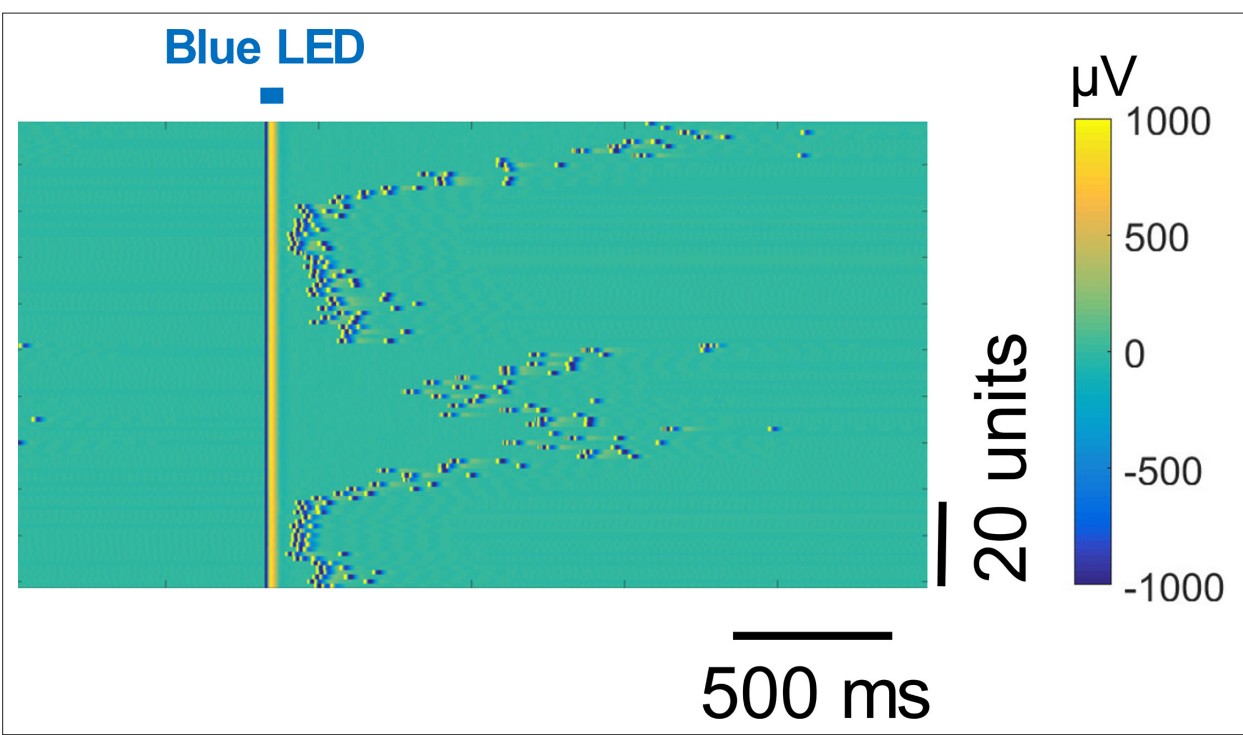

**Figure 7.** Light artefact in a recording from mouse prefrontal cortex showing that the application of a brief (1ms) pulse of light leads to saturation lasting about 5ms post-stimulation.

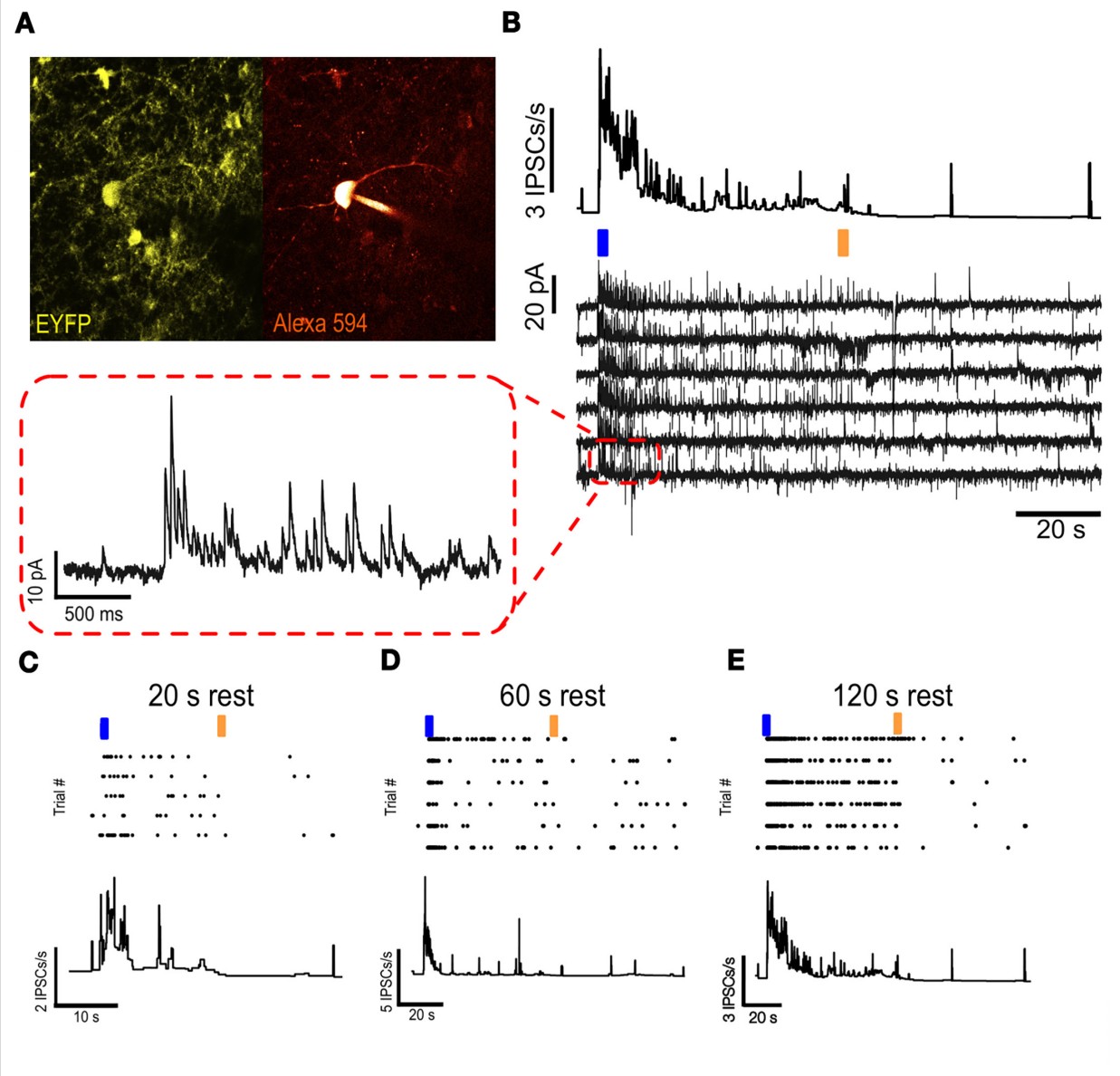

**Figure 8.** Validation of step-function opsin by whole-cell recordings. (**A**) Visually identified PV neuron (**B**) Nearby pyramidal neuron receiving inhibitory currents following PV opto activation (**C**) IPSCs onto pyramidal cell with 20 s rest between trial (**D**) IPSCs onto pyramidal cell with 60 s rest between trial (**E**) IPSCs onto pyramidal cell with 120 s rest between trial.

protocols, allowing us to selectively activate and inactivate PV interneurons in a temporally precise fashion using tools compatible with multielectrode array technology.

## Optogenetic validation of E/I classification on multielectrode arrays

To validate our prior waveform-based cell-type classification in mouse prefrontal cortex slices, we selectively activated transfected PV interneurons with the step-function opsin construct while performing multielectrode recordings (*Table 2*). To prevent the propagation of optogenetic activation throughout the network, we first recorded in the presence of 10 μM cyanquixaline (CNQX) and 100 μM picrotoxin (PTX) to block AMPA receptors and $GABA_A$ receptors, respectively (leading to a reduction in FR of 0.0937 Hz, $p < 0.001$ from baseline). We photostimulated with brief 100 ms pulses of blue light and examined post-stimulation responses of all identified neurons recorded (n=8189, 3 slices, from 3 mice; *Figure 9A*). In the presence of synaptic blockers, optogenetic activation of targeted PV interneurons led to a significant increase in mean firing rate for the population of cells that had been classified

**Table 2.** Summary of SpikeMAP open-source data.

Datasets were obtained from four mice. The optogenetic stimulation protocol (see Materials and methods) was run once per dataset. Data is available from: https://doi.org/10.6084/m9.figshare.29416472.

| File name | Animal # | Protocol # | # of E cells identified | # of I cells identified | Total recording time (min) |
|---|---|---|---|---|---|
| R20211221_Slice1_Vars3.mat | 1 | 1 | 3687 | 47 | 8 |
| R20211213_Slice2_Vars3.mat | 2 | 1 | 2943 | 350 | 8 |
| R20211209_Slice3_Vars3.mat | 3 | 1 | 1994 | 712 | 8 |
| R20211219_Slice3_Vars3.mat | 4 | 2 | 1138 | 32 | 8 |

as inhibitory based on their waveform properties (I neurons; *Figure 9B*, n=430, p<0.003). Neurons classified as excitatory neurons based on their waveform kinetics, on the other hand, showed only a subtle decrease in mean firing rate (*Figure 9B*, n=7759, p<0.001). A clustering analysis on the spike waveform properties of putative excitatory and inhibitory neurons provides further evidence of classification (*Figure 10*). The overall firing rates in slice recordings were relatively low and thus were not permissive to the use of cross-correlations in validating putative excitatory and inhibitory neurons. While our pharmacological manipulations attempted to largely eliminate network effects, the slight

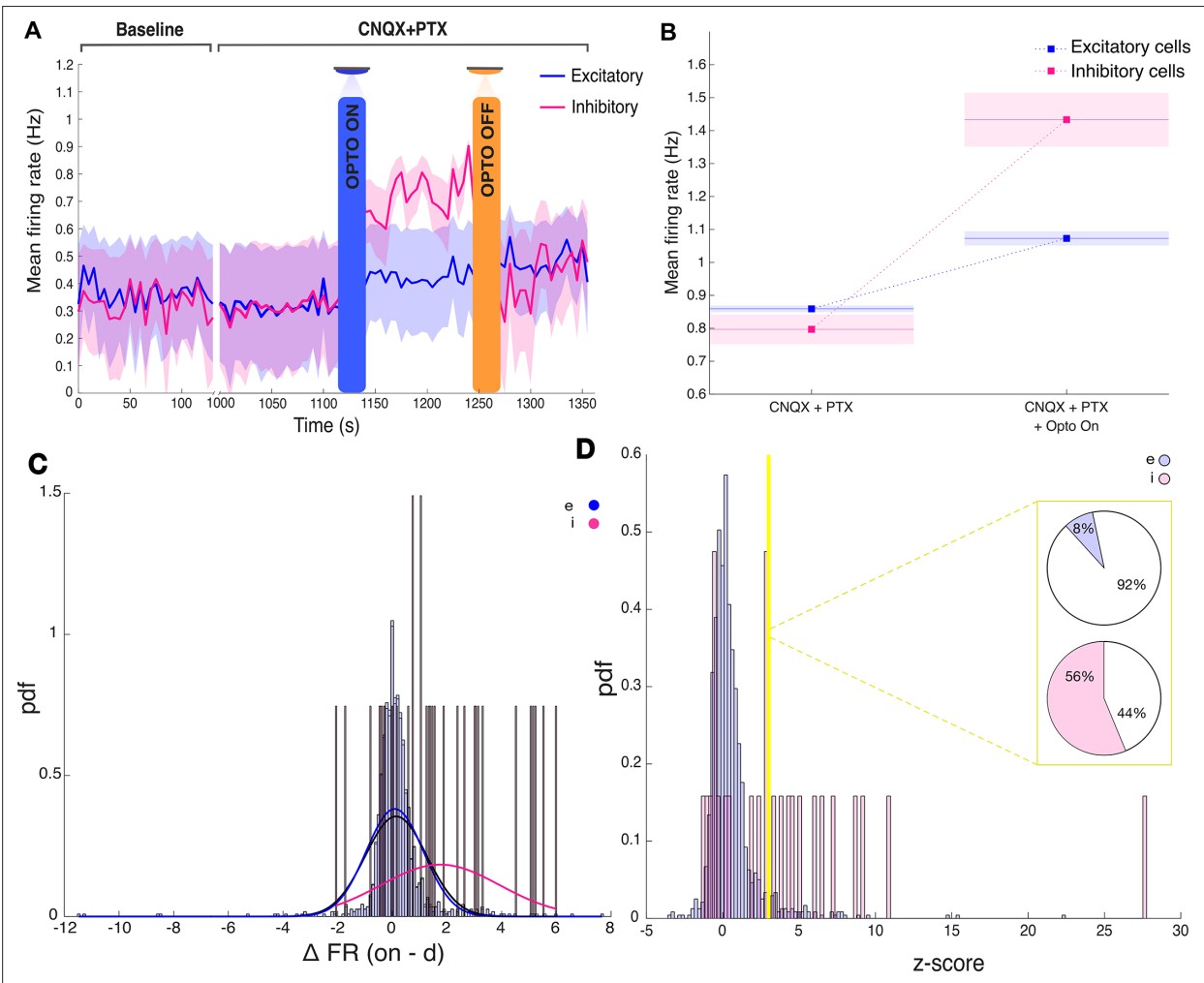

**Figure 9.** Optogenetic targeting of inhibitory cells for spike sorting (**A**) Average firing rate for putative excitatory and inhibitory neurons. The shaded area represents the SEM. (**B**) Quantification of change in firing rate following optogenetic stimulation. Average firing rates are taken over four recordings obtained from three mice. (**C**) Delta change in firing rate (on – drugs) (**D**) Opto-evoked change in firing rate.

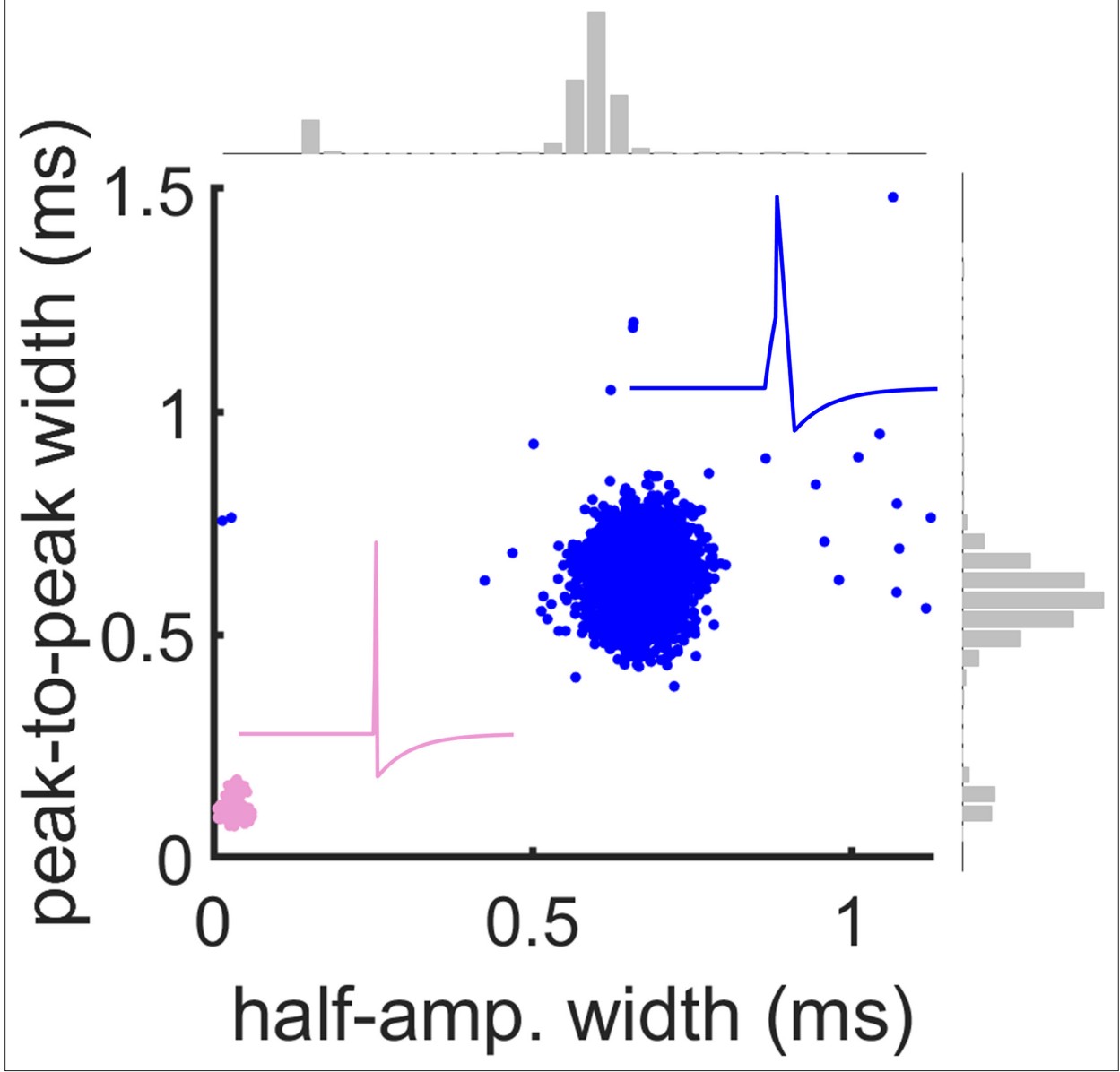

**Figure 10.** Waveform kinetics for E and I cells obtained during the baseline window of the optogenetic protocol. Insets show cartoons of individual spikes with broader (**E**) and narrower (**I**) waveforms.

reduction in firing rate of putative excitatory neurons may reflect the involvement of GABA$_B$ receptor-mediated inhibition (***Kohl and Paulsen, 2010***).

Next, we examined in further detail the response profiles of individual neurons to photostimulation in order to validate our algorithm at the single cell level. In the absence of synaptic blockers, individual neurons displayed a diverse range of responses to light stimulation, including neurons showing no significant change in firing rate ($\Delta$FR ~0), neurons showing an increase in firing rate ($\Delta$FR >0), and neurons showing a decrease in firing rate ($\Delta$FR <0; ***Figure 9C***). We applied a strict criterion to separate neurons showing a statistically robust increase in firing rate (see Materials and methods). We found that 56% of neurons that had been classified as I displayed a light-induced increase in firing rate. Conversely, we found that only 8% of neurons that had been classified as E showed an increase in firing rate after photostimulation. The percentage of I cells responsive to light may be influenced by our viral strategy, which targets only a subset of PV-expressing interneurons largely located in the infralimbic and prelimbic cortical subregions. Additionally, non-PV inhibitory interneurons and some excitatory neurons could have been initially classified as inhibitory based on their waveform properties. These

experiments, however, unambiguously identified a subset of genetically identified PV neurons: those that showed a transient photostimulation-induced increase in firing rate. Through a comparison with FWHM and PTP features, these results demonstrate that a step-function opsin approach is a viable and scalable method for validating spike waveform classification on multielectrode arrays.

Together, this provides evidence for cell classification of excitatory and inhibitory neurons based on their extracted waveform properties using spikeMAP, showing that cells identified as inhibitory displayed optogenetic-specific change in responses.

## Discussion

Extracellular electrophysiological recordings have allowed monitoring the spiking of thousands of neurons simultaneously. To meaningfully interpret these dynamics, spike sorting is a crucial step in the analysis of those recordings. The analysis of network dynamics can be severely impacted by improper interpretation of the underlying neural activity (*Rey et al., 2015*; *Carlson and Carin, 2019*), highlighting the need for optimal methods. With technological progress constantly allowing for denser arrays, assigning spikes to single cells and determining cell-type identity requires an automated solution given that the size of the data prevents manual curation. Most spike sorting methods are at least somewhat supervised (*Marre et al., 2012*; *Prentice et al., 2011*), which is not feasible on large-scale arrays. Here, we described an open-source, unsupervised, and scalable suite of computational tools that perform spike sorting and cell type classification on large-scale, high-density recordings. We integrated our analysis pipeline with an experimental protocol, allowing for the validation of our statistical method and for the optogenetic manipulation of distinct cell types in dense networks.

Previous work has detailed cell-type classification based on waveform analysis, but experimental validation has been lacking. To address this and provide biological evidence that this classification indeed correlates with cell type, we selectively activated PV interneurons using a combination of viral strategies and optogenetics while simultaneously recording from the whole network. Analysis of optogenetically stimulated trials revealed that cells that had been classified as inhibitory were more likely to display a change in firing rate during optogenetic stimulation.

We found that clustering in two dimensions, representing the full width of spikes at half-maximum amplitude and the peak-to-peak, achieved a high level of success in classifying cell types. By sampling electrodes from PV-positive transfected cells, we distinguished their action potentials from other cell types. Our work provides evidence that extracellular recordings can offer insights on the distinct neuronal populations that make up complex networks. Our pipeline encourages the integration of viral strategies, optogenetics, and electrophysiological recordings to manipulate complex networks to better understand the contribution of each cell type.

While SpikeMAP is the only known method to enable the identification of putative excitatory and inhibitory neurons on high-density multielectrode arrays (*Table 1*), several aspects of SpikeMAP included in the spike sorting pipeline (*Figure 1*) overlap with existing methods, as these constitute required steps prior to performing E/I identification. To enable users the ability to integrate SpikeMAP with existing toolboxes, we provide a modularized suite of protocols so that E/I identification can be performed separately from preliminary spike sorting steps. In this way, a user could carry out spike sorting through Kilosort or another package before importing their data to SpikeMAP for E/I identification.

Cortical inhibitory neurons are heterogeneous, and the fact that fast-spiking PV interneurons represent the largest population exhibiting landmark features of narrow-spike made them an attractive candidate to validate our classification. However, future work could include different inhibitory interneurons such as somatostatin (SOM) and vasoactive intestinal polypeptide (VIP) neurons to improve the classification of inhibitory cell types. Another avenue could involve applying SpikeMAP on artificially generated spike data (*Buccino and Einevoll, 2021*; *Laquitaine et al., 2024*). Nevertheless, it is possible that distinguishing all inhibitory cell types based on extracellular waveforms alone may not be achievable. Despite this limitation, incremental advancements such as those presented here are crucial for refining our understanding and improving analytical approaches. Given that spike sorting features in vitro can differ from properties exhibited in vivo (*Henze et al., 2000*; *Nowak et al., 2003*; *González-Burgos et al., 2005*), future work should address if these classification/waveform properties can be generalized to behaving organisms, for example with tetrodes, optrodes, or implanted electrode arrays.

In sum, our work described a fully integrated method with an open-source, unsupervised, and scalable spike sorting analysis pipeline performing cell-type classification, integrated with an experimental protocol utilizing a stabilized step function opsin to validate sorting on large-scale MEAs. These tools collectively provide a framework for investigating large-scale neural dynamics of genetically defined neural populations.

## Materials and methods

### Animals

PV-CRE mice (*Pvalb^Cre* gene designation, 008069) were acquired from Jackson Laboratory (Bar Harbor, ME, USA). All mouse lines were homozygous and in C57BL/6x129S4 background. Mice were kept on a 12:12 hr light/dark cycle, with access to food and water ad libitum. All experiments and procedures were performed in accordance with approved procedures and guidelines set forth by the University of Ottawa Animal Care and Veterinary Services (protocol # 3471).

### Stereotaxic injections

For stereotaxic injections, P21 mice were injected with 0.05 mg/kg buprenorphine and anesthetized by inhalation of isoflurane. Injections were performed using a 10 μL Hamilton syringe with a 33-gauge needle, and 1 μL was injected per coordinate. Stereotaxic coordinates are as follows, from the bregma landmark: mPFC [anterior–posterior (AP), +2.4 —> +2.6; medial–lateral (ML), ±0.42; dorsal–ventral (DV), –1.5 —>–1.8]. The DV coordinate was calculated from the surface of the pia. For ChR2 expression, 400 nL of pAAV-Ef1a-DIO hChR2 (C128S/D156A)-EYFP was injected, per site, (bilaterally) and animals were left to recover for 2–3 weeks before ex vivo electrophysiological recordings.

### Slices

mPFC slices were prepared from 35- to 42-day-old PV-CRE mice, 2–3 weeks post-stereotaxic injections. For slice preparation, mice were anesthetized by inhalation of isoflurane (Baxter Corporation) and killed by decapitation. The brain was removed, and coronal brain slices were sectioned while immersed in an ice-cold choline chloride-based cutting solution of the following composition: 119 mM choline-Cl, 2.5 mM KCl, 1 mM CaCl2, 4.3 mM MgSO4-7H2O, 1 mM NaH2PO4, 1.3 mM sodium L-ascorbate, 26.2 mM NaHCO3, and 11 mM glucose, and equilibrated with 95% $O_2$ and 5% $CO_2$. Slices were recovered in a chamber containing standard artificial cerebrospinal fluid solution (aCSF) of the following composition: 119 mM NaCl, 2.5 mM CaCl2, 1.3 mM MgSO4- 7H2O, 1 mM NaH2PO4, 26.2 mM NaHCO3, and 11 mM glucose, at a temperature of 37 °C, continuously bubbled with a mixture of 95% O2 and 5% CO2. Slices were recovered for 1 hr in the recovery chamber and equilibrated to a temperature of ~25 °C before the recordings were performed. The tissue was continuously perfused with fresh aCSF. The recordings shown in this paper were 24 min long and were as follows: baseline (2 min), opto-on (2 min), opto-off (2 min), CNQX wash-on (5 min), CNQX +opto on (2 min), CNQX +opto off (2 min), PTX wash-on (5 min), CNQX and PTX +opto on (2 min), CNQX and PTX +opto off (2 min). MEA-chip baseline (no slice) was recorded (30 s) as control prior to adding each slice. Slices were imaged on the multi-electrode array and preserved for immunohistochemistry verification of virus expression site.

### Whole-cell electrophysiology

Neurons were visualized for whole-cell recordings using an upright microscope: BX61WU upright microscope (60 X, 1.0 NA objective; Olympus, Melville, NY). PV neurons were visually identified by EYFP expression. Whole-cell recordings were performed using borosilicate glass patch electrodes (3–6 MΩ; World Precision Instruments) pulled on a Narishige PC-10 pipette puller. Electrical signals were recorded using an Axon Multiclamp 700B amplifier, filtered at 2 kHz, and digitized at 10 kHz with an Axon Digidata 1440 A digitizer. All experiments were performed at room temperature in Ringer's solution containing (in mM) 119 NaCl, 2.5 CaCl2, 1.3 MgSO4-7H20, 1 NaH2PO4, and 26.2 NaHCO3, and 11 glucose saturated with 95% O2 and 5% CO2 (pH 7.3, 295–310 mOsm/L). Electrodes were filled with an intracellular solution of the following compositions (in mM): 115 K-gluconate, 20 KCl, 10 HEPES, 4 ATP-Mg, 0.5 GTP, 10 Na-phosphocreatine (pH = 7.2–7.3; 270–290 mOsm/L).

## Multielectrode array recordings

Extracellular voltage was recorded using High-Density Multi Electrode Arrays (HD-MEAs) from 3 Brain, Switzerland (https://www.3brain.com/). A detailed description of the technology behind this system can be found in *Imfeld et al., 2008* and in *Berdondini et al., 2009*. High-density recordings from the mouse's prefrontal cortex were performed in vitro using the BioCAM X platform with HD-MEA Arena chips (3Brain GmbH, Switzerland). This system records using 4096 square microelectrodes (pitch = 42 μm) positioned in a 64x64 layout (2.67mm x 2.67 mm area) with a sampling rate fs of 18 kHz when recording from the entire 64x64 array. Activity was recorded at 12 bits resolution per channel. The data was visualized and acquired with the BrainWaveX software provided by 3Brain.

## Spike sorting

A code library for SpikeMAP in Python and Matlab is available as open source software from figshare, as well as from GitHub (copy archived at *Giraud, 2020*).

## Filtering

The extracellular voltages recorded by multielectrode arrays typically reflect the dynamics of multiple cells in proximity to a given electrode (*Rey et al., 2015*), raising the problem of how to isolate individual units. The unfiltered voltage at individual channels was characterized by sharp deflections (*Figure 2Bi*) that are typical of in vitro extracellular activity in spontaneously active cortical circuits (*Hemberger et al., 2019*). As a necessary first step to isolate individual units, we filtered the extracellular voltage using a second-order Butterworth bandpass filter (300–5000 Hz; *Figure 2Bii*) to remove slowly changing field potentials and high-frequency noise (*Bullmann et al., 2019*). Recording artefacts (voltage deflection > ± 500 μV) were removed and set to the mean voltage.

## Spike detection

Putative spike times were detected using a voltage-threshold method (*Lewicki, 1998*), identifying negative signal peaks below a threshold of $x$ standard deviation (SD) from the mean of the filtered signal. Here, $x$ is calculated for each recording, by plotting the filtered voltage histogram for each channel and fitting it with a Gaussian curve to determine the SD value at which non-normal, putative spike-related activity is found. An absolute refractory period was applied, and successive events within <2ms were discarded. Next, the unfiltered voltage around the time of each spike (from –5 to +5ms) was extracted (*Figure 2Bii*), yielding a series of high-frequency (18 kHz) waveforms for each channel. For the data shown here, all three recordings used had $x = 4.5$ SD.

## Isolation of single units

To isolate single-cell activity, we employed a principal component analysis on individual channels, which revealed distinct clusters putatively corresponding to individual cells (*Figure 2E*). Next, clusters amongst the principal components of rrSVD were extracted using a k-means cluster analysis (k=2; *Xu and Wunsch, 2005*). A linear discriminant analysis (LDA) was then applied to the labeled principal components (*Hill et al., 2011*) to determine how well the data were separated by this approach. LDA provides an estimate of linear classification error between the two clusters assuming equal class covariance. Tenfold cross-validation was employed, where chance corresponds to an error estimate of 0.5 (*Figure 2G*). Clusters with error lower than chance were merged by applying the same label to all waveforms. Each cluster was considered a putative neuron.

## Center-of-mass triangulation

In order to precisely estimate soma locations between each recording electrode, we used a center-of-mass analysis (*Muthmann et al., 2015*; *Hilgen et al., 2017*). For each putative soma identified previously, a 3-by-3 matrix with elements $a_{ij}$ was constructed by extracting the waveforms on adjacent electrodes. Location (2,2) of this matrix was the amplitude of the negative peak in mean waveform of the somatic electrode. Surrounding locations were populated with the amplitude of the negative peaks of all electrodes immediately adjacent to the somatic electrode. The row of the center-of-mass is given by

$$r = \frac{\sum_i \sum_j (i \cdot a_{ij})}{\sum_{ij} a_{ij}}$$

and the column is

$$c = \frac{\sum_i \sum_j (j \cdot a_{ij})}{\sum_{ij} a_{ij}}.$$

In addition to center-of-mass triangulation, SpikeMAP includes protocols to perform monopolar triangulation and grid-based convolution, offering additional options to estimate putative soma locations based on waveform amplitudes.

## Spline interpolation

The mean waveform for each single cell was resampled at a higher rate using a piecewise cubic spline interpolation (Matlab R2015b), constructing new points within the boundaries of a set of known points using query points at 90 kHz.

## E/I classification

We used a combination of cell-type-specific viral strategies and optogenetics to address cell-type distinction. We injected PV-CRE mice (B6.Cg-Pvalb<tm1.1(cre)Aibs>/J) with a stabilized step-function opsin (pAAV-Ef1a-DIO hChR2 (C128S/D156A)-EYFP) bilaterally in the medial prefrontal cortex (mPFC; *Berndt et al., 2009*). This allows for selective, prolonged activation of parvalbumin (PV) interneurons with a 100ms pulse of 465 nm blue light and deactivation with a 10 s pulse of 545–580 nm green-yellow light. Next, we computed a z-score for each neuron corresponding to photostimulation-evoked change in firing rate, expressed in units of SD from the baseline mean (*Figure 5D*). We isolated cells showing a robust increase in firing rate (z score >3). A total of (n=8189, 3 slices, from 3 mice) was classified in this fashion. Overall, 7759 (94.75 %) were reliably classified as excitatory and 430 (5.25 %) were reliably identified as inhibitory. This number is close to the expected percentage of PV interneurons in cortex (4–6%; *Markram et al., 2004*).

## LED photostimulation

LED photostimulation was delivered using a PlexBright LED module and controller (465 nm; Plexon). Light was delivered through a 200 μm patch fiber cable with a bare fiber tip with ~1 cm of glass exposed. Photodeactivation was delivered using a yellow activation wideband 545–580 nm filter (BP545-580; Olympus).

## Immunohistochemistry

Brain slices used for immunohistochemistry were prepared from 35- to 42-day-old mice with 1×PBS used as the solvent for all solutions. Mice were anesthetized by inhalation of isoflurane. They were then intracardially perfused with 5 mL of PBS pH = 7.4 followed by 10 mL of 4% paraformaldehyde/ 0.1 M PBS, pH 7.4 for 10 min. Postfixation in 4% PFA was carried out for 2 hr before sequential cryoprotection steps in 15% and 30% sucrose over the course of 2 d. Brains were frozen in −30 °C isopentane, mounted in cryomatrix, and cut to 40 μm slices on a cryostat (CM 3050 S; Leica). Free-floating sections were blocked in 0.5% PBS-Triton-X-100 (TX-100) for 45 min and incubated in primary antibody against GFP (GFP-1020; 1:1000, AvesLabs) in blocking solution at 20 °C. On the following morning, slices were washed three times, for 5 min each time, with 0.5% PBS-Triton-X-100 before incubation in the secondary antibody goat anti-chicken–conjugated AlexaFluor 488 (Jackson Labs, 703-546-155) for 3 hr at room temperature. Slices were washed with 0.1% Tx-100 three times, for 5 min each time, before mounting onto slides with Vectashield (Vector Labs). Additionally, all slices used for recordings were saved, kept immersed in Formalin 10% in PBS pH = 7.4, for immunohistochemistry to verify viral expression. The protocol is the same as above.

## Acknowledgements

This work was supported by a discovery grant from the Natural Sciences and Engineering Council of Canada (NSERC grants 210977; 210989; 2020–06901), the Canadian Institute for Health Research

(175325) by the Fonds de recherche du Québec in Nature and technologie (doctoral scholarship), by the Natural Sciences and Engineering Council of Canada (NSERC CGS M scholarship) and the University of Ottawa Brain and Mind Institute.

## Additional information

### Funding

| Funder | Grant reference number | Author |
| --- | --- | --- |
| Natural Sciences and Engineering Research Council of Canada | 210977 | Jean-Philippe Thivierge |
| Natural Sciences and Engineering Research Council of Canada | 210989 | Jean-Claude Beique |
| Natural Sciences and Engineering Research Council of Canada | 2020-06901 | Jean-Claude Beique |
| Canadian Institutes of Health Research | 175325 | Jean-Claude Beique |
| Fonds de recherche du Québec – Nature et technologies | Doctoral scholarship | Eloise Giraud |
| Natural Sciences and Engineering Research Council of Canada | CGS M scholarship | Eloise Giraud |
| University of Ottawa Brain and Mind Institute | | Jean-Claude Beique Jean-Philippe Thivierge |

The funders had no role in study design, data collection and interpretation, or the decision to submit the work for publication.

### Author contributions

Eloise Giraud, Conceptualization, Resources, Data curation, Software, Formal analysis, Validation, Investigation, Visualization, Methodology, Writing – original draft, Writing – review and editing; Michael Lynn, Software, Formal analysis, Methodology, Writing – review and editing; Philippe Vincent-Lamarre, Data curation, Software, Formal analysis, Validation, Methodology, Writing – review and editing; Jean-Claude Beique, Resources, Supervision, Funding acquisition, Project administration, Writing – review and editing; Jean-Philippe Thivierge, Conceptualization, Resources, Data curation, Software, Formal analysis, Supervision, Funding acquisition, Validation, Investigation, Visualization, Methodology, Writing – original draft, Project administration, Writing – review and editing

### Author ORCIDs

Jean-Claude Beique ⓘ https://orcid.org/0000-0001-7278-4906
Jean-Philippe Thivierge ⓘ https://orcid.org/0000-0003-2457-7173

### Ethics

All experiments and procedures were performed in accordance with approved procedures and guidelines set forth by the University of Ottawa Animal Care and Veterinary Services (protocol # 3471).

Reviewer #1 (Public review): https://doi.org/10.7554/eLife.106557.3.sa1
Reviewer #2 (Public review): https://doi.org/10.7554/eLife.106557.3.sa2
Author response https://doi.org/10.7554/eLife.106557.3.sa3

## Additional files

### Supplementary files
MDAR checklist

### Data availability
All data generated or analysed during this study are included in the manuscript, supporting files, and supplementary data available online.

The following datasets were generated:

| Author(s) | Year | Dataset title | Dataset URL | Database and Identifier |
|---|---|---|---|---|
| Thivierge JP | 2025 | SpikeMAP: An unsupervised spike sorting pipeline for cortical excitatory and inhibitory neurons in high-density multielectrode arrays | https://doi.org/10.6084/m9.figshare.28458749.v1 | figshare, 10.6084/m9.figshare.28458749.v1 |
| Thivierge JP | 2025 | Datasets of high-density multielectrode recordings obtained from four mice | https://doi.org/10.6084/m9.figshare.29416472 | figshare, 10.6084/m9.figshare.29416472 |

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
