## [Editor Report · eLife Assessment]

In this manuscript, the authors describe a software package for automatic differentiation of action potentials generated by excitatory and inhibitory neurons, acquired using high-density microelectrode arrays. The work is **valuable** as it offers a tool with the potential to automatically identify these neuron types in vitro. It is **solid**, as it provides a tool to identify putative excitatory and inhibitory neurons on high-density electrode arrays, which can be used in conjunction with other existing spike sorting pipelines.

---

## [Referee Report · Reviewer #1 (Public review)]

Summary:

The authors note that while many software packages exist for spike sorting, these do not automatically differentiate with known accuracy between excitatory and inhibitory neurons. Moreover, most existing spike sorting packages are for in vivo use, where the majority of electrodes are separated from each other by several hundred microns or more. There is a need for spike sorting packages that can take advantage of high-density electrode arrays where all electrodes are within a few tens of microns from other electrodes. Here, the authors offer such a software package with SpikeMAP, and they validate its performance in identifying parvalbumin interneurons that were optogenetically stimulated.

Strengths:

The main strength of this work is that the authors use ground truth measures to show that SpikeMAP can take features of spike shapes to correctly identify known parvalbumin interneurons against a background of other neuron types. They use spike width and peak to peak distance as the key features for distinguishing between neuron types, a method that has been around for many years (Barthó, Peter, et al. "Characterization of neocortical principal cells and interneurons by network interactions and extracellular features." Journal of neurophysiology 92.1 (2004): 600-608.), but whose performance has not been validated in the context of high-density electrode arrays.

Another strength of this approach is that it is automated - a necessity if your electrode array has 4096 electrodes. Hand-sorting or even checking such a large number of channels is something even the cruellest advisor would not wish upon a graduate student. With such large channel counts, it is essential to have automated methods that are known to work accurately. Hence, the combination of validation and automation is an important advance.

A nice feature of this work is that with high-density electrode arrays, the spike waveforms appear on multiple nearby electrodes simultaneously. And since spike amplitudes fall off with distance, this allows triangulation of neuron locations within the regular electrode array. Thus, spike correlations between neuron types, or within neuron types, can be plotted as a function of distance. While SpikeMAP is not the first to do this (Peyrache, Adrien, et al. "Spatiotemporal dynamics of neocortical excitation and inhibition during human sleep." Proceedings of the national academy of sciences 109.5 (2012): 1731-1736.), it is a welcome capability of this package.

It is also good that the code for this package is open-source, allowing a community of people (I expect in vitro labs will especially want to use this) to use the code and further improve it.

Weaknesses:

As this code was developed for use with a 4096-electrode array, it is important to be aware of double counting neurons across the many electrodes. I understand that there are ways within the code to ensure that this does not happen, but care must be taken in two key areas: First, action potentials traveling down axons will exhibit a triphasic waveform that is different from the biphasic waveform that appears near the cell body, but these two signals will still be from the same neuron (for example, see Litke et al., 2004 "What does the eye tell the brain: Development of a System for the Large-Scale Recording of Retinal Output Activity"; figure 14). I did not see anything that would directly address this situation, so it might be something for you to consider in updated versions of the code. Second, spike shapes are known to change when firing rates are high, like in bursting neurons (Harris, K.D., Hirase, H., Leinekugel, X., Henze, D.A. & Buzsáki, G. Temporal interaction between single spikes and complex spike bursts in hippocampal pyramidal cells. Neuron 32, 141-149 (2001)). I did not see this addressed in the present version of the manuscript.

Another area for possible improvement would be to build on the excellent validation experiments you have already conducted with parvalbumin interneurons. Although it would take more work, similar experiments could be conducted for somatostatin and vasoactive intestinal peptide neurons against a background of excitatory neurons. These may have different spike profiles, but your success in distinguishing them can only be known if you validate against ground truth, like you did for the PV interneurons.

Appraisal:

This work addresses the need for an automated spike sorting software package for high density electrode arrays. Although no spike sorting software is flawless, the package presented here, SpikeMAP, has been validated on PV interneurons, inspiring a degree of confidence. This is a good start, and further validation on other neuron types could increase that confidence. Groups doing in vitro experiments, where 4096 electrode arrays are more common, could find this system particularly helpful.

Comments on revised version:

I appreciate the dialogue that has occurred over this submission. I have seen how the authors have taken into account the issues that I have raised, as well as those brought up by reviewer 2. I am satisfied that the paper has improved and is now a novel and useful contribution in the area of spike sorting.

---

## [Referee Report · Reviewer #2 (Public review)]

Summary:

In this paper, entitled "SpikeMAP: An unsupervised spike sorting pipeline for cortical excitatory and inhibitory 2 neurons in high-density multielectrode arrays with ground-truth validation", the authors are presenting spikeMAP, a pipeline for the analysis of large-scale recordings of in vitro cortical activity. According to the authors, spikeMAP not only allows for the detection of spikes produced by single neurons (spike sorting), but also allows for the reliable distinction between genetically determined cell types by utilizing viral and optogenetic strategies as ground-truth validation. While I find that the paper is nicely written, and easy to follow, I find that the algorithmic part of the paper is not really new and should have been more carefully compared to existing solutions. While the GT recordings to assess the possibilities of a spike sorting tool to distinguish properly between excitatory and inhibitory neurons is interesting, spikeMAP does not seem to bring anything new to state of the art solutions, and/or, at least, it would deserve to be properly benchmarked. This is why I would suggest the authors to perform a more intensive comparison with existing spike sorters.

Strengths:

The GT recordings with optogenetic activation of the cells, based on the opsins is interesting and might provide useful data to quantify how good spike sorting pipelines are, in vitro, to discriminate between excitatory and inhibitory neurons. Such an approach can be quite complementary with artificially generated ground truth.

Weaknesses:

The global workflow of spikeMAP, described in Figure 1, seems to be very similar to the one of [Hilgen et al, 2020, 10.1016/j.celrep.2017.02.038.]. Therefore, the first question is what is the rationale of reinventing the wheel, and not using tools that are doing something very similar (as mentioned by the authors themselves). I have a hard time, in general, believing that spikeMAP has something particularly special, given its Methods, compared to state-of-the-art spike sorters. This is why at the very least, the title of the paper is misleading, because it let the reader think that the core of the paper will be about a new spike sorting pipeline. If this is the main message the authors want to convey, then I think that numerous validations/benchmarks are missing to assess first how good spikeMAP is, w.r.t. spike sorting in general, before deciding if this is indeed the right tool to discriminate excitatory vs inhibitory cells. The GT validation, while interesting, is not enough to entirely validate the paper. The details are a bit too scarce to me, or would deserve to be better explained (see other comments after)

Regarding the putative location of the spikes, it has been shown that center of mass, while easy to compute, is not the most accurate solution [Scopin et al, 2024, 10.1016/j.jneumeth.2024.110297]. For example, it has an intrinsic bias for finding positions within the boundaries of the electrodes, while some other methods such as monopolar triangulation or grid-based convolution might have better performances. Can the authors comment on the choice of Center of Mass as a unique way to triangulate the sources?

Still in Figure 1, I am not sure to really see the point of Spline Interpolation. I see the point of such a smoothing, but the authors should demonstrate that it has a key impact on the distinction of Excitatory vs. Inhibitory cells. What's special with the value of 90kHz for a signal recorded at 18kHz? What is the gain with spline enhancement compared to without? Does such a value depend on the sampling rate, or is it a global optimum found by the authors?

Figure 2 is not really clear, especially panel B. The choice of the time scale for the B panel might not be the most appropriate, and the legend filtered/unfiltered with a dot is not clear to me in Bii. In panel E, the authors are making two clusters with PCA projections on single waveforms. Does this mean that the PCA is only applied to the main waveforms, i.e. the ones obtained where the amplitudes are peaking the most? This is not really clear from the methods, but if this is the case, then this approach is a bit simplistic and not really matching state-of-the-art solutions. Spike waveforms are quite often, especially with such high-density arrays, covering multiple channels at once and thus the extracellular patterns triggered by the single units on the MEA are spatio-temporal motifs occurring on several channels. This is why, in modern spike sorters, the information in a local neighbourhood is often kept to be projected, via PCA, on the lower dimensional space before clustering. Information on a single channel only might not be informative enough to disambiguate sources. Can the authors comment on that, and what is the exact spatial resolution of the 3Brain device? The way the authors are performing the SVD should be clarified in the methods section. Is it on a single channel, and/or on multiple channels in a local neighbourhood?

About the isolation of the single units, here again, I think the manuscript lacks some technical details. The authors are saying that they are using a k-means cluster analysis with k=2. This means that the authors are explicitly looking for 2 clusters per electrodes. If so, this is a really strong assumption that should not be held in the context of spike sorting, because since it is a blind source separation technique, one cannot pre-determine in advance how many sources are present in the vicinity of a given electrode. While the illustration on Figure 2E is ok, there is no guarantee that one cannot find more clusters, so why this choice of k=2? Again, this is why most modern spike sorting pipelines are not relying on k-means, to avoid any hard coded number of clusters. Can the authors comment on that?

I'm surprised by the linear decay of the maximal amplitude as a function of the distance from soma, as shown in Figure 2H. Is it really what should be expected? Based on the properties of the extracellular media, shouldn't we expect a power law for the decay of the amplitude? This is strange that up to 100um away from the some, the max amplitude only dropped from 260 to 240 uV. Can the authors comment on that? It would be interesting to plot that for all neurons recorded, in a normed manner V/max(V) as function of distances, to see what the curve looks like

In Figure 3A, it seems that the total number of cells is rather low for such a large number of electrodes. What are the quality criteria that are used to keep these cells? Did the authors exclude some cells from the analysis, and if yes, what are the quality criteria that are used to keep cells? If no criteria are used (because none is mentioned in the Methods), then how come so few cells are detected, and can the authors convince us that these neurons are indeed "clean" units (RPVs, SNRs, ...)

Still in Figure 3A, it looks like there is a bias to find inhibitory cells at the borders, since they do not appear to be uniformly distributed over the MEA. Can the authors comment on that? What would be the explanation for such a behaviour? It would be interesting to see some macroscopic quantities on Excitatory/Inhibitory cells, such as mean firing rates, averaged SNRs, ... Because again, in Figure 3C, it is not clear to me that the firing rates of inhibitory cells is higher than Excitatory ones, while it should be in theory.

For Figure 3 in general, I would have performed an exhaustive comparison of putative cells found by spikeMAP and other sorters. More precisely, I think that to prove the point that spikeMAP is indeed bringing something new to the field of spike sorting, the authors should have compared the performances of various spike sorters to discriminate Exc vs Inh cells based on their ground truth recordings. For example, either using Kilosort [Pachitariu et al, 2024, 10.1038/s41592-024-02232-7], or some other sorters that might be working with such large high-density data [Yger et al, 2018, 10.7554/eLife.34518]

Figure 4 has a big issue, and I guess the panels A and B should be redrawn. I don't understand what the red rectangle is displaying.

I understand that Figure 4 is only one example, but I have a hard time understanding from the manuscript how many slices/mice were used to obtain the GT data? I guess the manuscript could be enhanced by turning the data into an open access dataset, but then some clarification is needed. How many flashes/animals/slices are we talking about. Maybe this should be illustrated in Figure 4, if this figure is devoted to the introduction of the GT data.

While there is no doubt that GT data as the ones recorded here by the authors are the most interesting data from a validation point of view, the pretty low yield of such experiments should not discourage the use of artificially generated recordings such as the ones made in [Buccino et al, 2020, 10.1007/s12021-020-09467-7] or even recently in [Laquitaine et al, 2024, 10.1101/2024.12.04.626805v1]. In these papers, the authors have putative waveforms/firing rates patterns for excitatory and inhibitory cells, and thus the authors could test how good they are in discriminating the two subtypes

Comments on revised version:

While I must thank the authors for their answers, I still think that they miss an important one, and only partially answering some of my concerns.

I truly think that SpikeMAP would benefit with a comparison with a state-of-the-art spike sorting pipeline, for example Kilosort. The authors said that they made the sorter modular enough such that only the E/I classification step can be compared. I think this would be worth it, just to be sure that SpikeMAP spike sorting, which might be more simple than other recent solution (with template matching), is not missing some cells, and thus degrading the E/I classification performances. I know that such a comparison is not straightforward, because there is no clear ground truth, but I would still need to be convinced that the sorting pipelines is bringing something, on its own. While there is no doubt that the E/I classification layer can be interesting, especially given the recordings shared by the authors, I'm still a bit puzzled by the sorting step. Thus maybe either a Table, a figure, or even as Supplementary one. Or the authors could try to generate fake GT data with MEArec for example, with putative E/I cells (discriminated via waveforms and firing rates) and show on such (oversimplified) data that SpikeMAP is performing similarly to modern spike sorters. Otherwise, this is a bit hard to judge...

---

## [Author Response]

The following is the authors’ response to the original reviews.

**Reviewer #1 (Public review)**
As this code was developed for use with a 4096 electrode array, it is important to be aware of double-counting neurons across the many electrodes. I understand that there are ways within the code to ensure that this does not happen, but care must be taken in two key areas. Firstly, action potentials traveling down axons will exhibit a triphasic waveform that is different from the biphasic waveform that appears near the cell body, but these two signals will still be from the same neuron (for example, see Litke et al., 2004 "What does the eye tell the brain: Development of a System for the Large-Scale Recording of Retinal Output Activity"; figure 14). I did not see anything that would directly address this situation, so it might be something for you to consider in updated versions of the code.

Thank you for this comment. We have added a routine to the SpikeMAP to remove highly correlated spikes detected within a given spatial radius of each other. The following was added to the main text (line 149):

“As an additional verification step, SpikeMAP allows the computation of spike-count correlations between putative neurons located within a user-defined radius. Signals that exceed a defined threshold of correlation can be rejected as they likely reflect the same underlying cell.”

Secondly, spike shapes are known to change when firing rates are high, like in bursting neurons (Harris, K.D., Hirase, H., Leinekugel, X., Henze, D.A. & Buzsáki, G. Temporal interaction between single spikes and complex spike bursts in hippocampal pyramidal cells. Neuron 32, 141-149 (2001)). I did not see this addressed in the present version of the manuscript.

We have added a routine to SpikeMAP that computes population spike rates to verify stationarity over time. We have also added a routine to identify putative bursting neurons through a Hartigan statistical dip test applied to the inter-spike distribution of individual cells.

We added the following (line 204):

“Further, SpikeMAP contains a routine to perform a Hartigan statistical dip test on the inter-spike distribution of individual cells to detect putative bursting neurons.”

Another area for possible improvement would be to build on the excellent validation experiments you have already conducted with parvalbumin interneurons. Although it would take more work, similar experiments could be conducted for somatostatin and vasoactive intestinal peptide neurons against a background of excitatory neurons. These may have different spike profiles, but your success in distinguishing them can only be known if you validate against ground truth, like you did for the PV interneurons.

We have added the following (line 326):

“future work could include different inhibitory interneurons such as somatostatin (SOM) and vasoactive intestinal polypeptide (VIP) neurons to improve the classification of inhibitory cell types. Another avenue could involve applying SpikeMAP on artificially generated spike data (Buccino & Einevoll 2021; Laquitaine et al., 2024).”

**Reviewer #2 (Public review)**
Summary:While I find that the paper is nicely written and easy to follow, I find that the algorithmic part of the paper is not really new and should have been more carefully compared to existing solutions. While the GT recordings to assess the possibilities of a spike sorting tool to distinguish properly between excitatory and inhibitory neurons are interesting, spikeMAP does not seem to bring anything new to state-of-the-art solutions, and/or, at least, it would deserve to be properly benchmarked. I would suggest that the authors perform a more intensive comparison with existing spike sorters.

Thank you for your insightful comment. A full comparison between SpikeMAP and related methods is provided in Table. 1. As can be seen, SpikeMAP is the only method listed that performs E/I sorting on large-scale multielectrodes. Nonetheless, several aspects of SpikeMAP included in the spike sorting pipeline do overlap with existing methods, as these constitute necessary steps prior to performing E/I identification. These steps are not novel to the current work, nor do they constitute rigid options that cannot be substituted by the user. Rather, we aim to offer SpikeMAP users the option to combine E/I identification with preliminary steps performed either through our software or through another package of their choosing. For instance, preliminary spike sorting could be done through Kilosort before importing the spike data into SpikeMAP for E/I identification. To allow greater flexibility, we have now modularized our suite so that E/I identification can be performed as a stand-alone module. We have clarified the text accordingly (line 317):

“While SpikeMAP is the only known method to enable the identification of putative excitatory and inhibitory neurons on high-density multielectrode arrays (Table 1), several aspects of SpikeMAP included in the spike sorting pipeline (Figure 1) overlap with existing methods, as these constitute required steps prior to performing E/I identification. To enable users the ability to integrate SpikeMAP with existing toolboxes, we provide a modularized suite of protocols so that E/I identification can be performed separately from preliminary spike sorting steps. In this way, a user could carry out spike sorting through Kilosort or another package before importing their data to SpikeMAP for E/I identification.”

Weaknesses:(1) The global workflow of spikeMAP, described in Figure 1, seems to be very similar to that of Hilgen et al. 2020 (10.1016/j.celrep.2017.02.038). Therefore, the first question is what is the rationale of reinventing the wheel, and not using tools that are doing something very similar (as mentioned by the authors themselves). I have a hard time, in general, believing that spikeMAP has something particularly special, given its Methods, compared to state-of-the-art spike sorters.

The paper by Hilgen et al. is reported in Table 1. As seen, while this paper employs optogenetics, it does not target inhibitory (e.g., PV) cells. We have added the following clarification (line 82):

“Despite evidence showing differences in action potential kinetics for distinct cell-types as well as the use of optogenetics (Hilgen et al., 2017), there exists no large-scale validation efforts, to our knowledge, showing that extracellular waveforms can be used to reliably distinguish cell-types.”

This is why, at the very least, the title of the paper is misleading, because it lets the reader think that the core of the paper will be about a new spike sorting pipeline. If this is the main message the authors want to convey, then I think that numerous validations/benchmarks are missing to assess first how good spikeMAP is, with reference to spike sorting in general, before deciding if this is indeed the right tool to discriminate excitatory vs inhibitory cells. The GT validation, while interesting, is not enough to entirely validate the paper. The details are a bit too scarce for me, or would deserve to be better explained (see other comments after).

We thank the reviewer for this comment, and have amended the title as follows:

“SpikeMAP: An unsupervised pipeline for the identification of cortical excitatory and inhibitory neurons in high-density multielectrode arrays with ground-truth validation”

(2) Regarding the putative location of the spikes, it has been shown that the center of mass, while easy to compute, is not the most accurate solution [Scopin et al, 2024, 10.1016/j.jneumeth.2024.110297]. For example, it has an intrinsic bias for finding positions within the boundaries of the electrodes, while some other methods, such as monopolar triangulation or grid-based convolution,n might have better performances. Can the authors comment on the choice of the Center of Mass as a unique way to triangulate the sources?

We agree with the reviewer that the center-of-mass algorithm carries limitations that are addressed by other methods. To address this issue, we have included two additional protocols in SpikeMAP to perform monopolar triangulation and grid-based convolution, offering additional options for users of the package. The text has been clarified as follows (line 429):

“In addition to center-of-mass triangulation, SpikeMAP includes protocols to perform monopolar triangulation and grid-based convolution, offering additional options to estimate putative soma locations based on waveform amplitudes.”

(3) Still in Figure 1, I am not sure I really see the point of Spline Interpolation. I see the point of such a smoothing, but the authors should demonstrate that it has a key impact on the distinction of Excitatory vs. Inhibitory cells. What is special about the value of 90kHz for a signal recorded at 18kHz? What is the gain with spline enhancement compared to without? Does such a value depend on the sampling rate, or is it a global optimum found by the authors?

We clarified the text as follows (line 183):

“While we found that a resolution of 90 kHZ provided a reasonable estimate of spike waveforms, this value can be adjusted as a parameter in SpikeMAP.”

(4) Figure 2 is not really clear, especially panel B. The choice of the time scale for the B panel might not be the most appropriate, and the legend filtered/unfiltered with a dot is not clear to me in Bii.

We apologize for the rendering issues in the Figures that occurred during conversion into PDF format. We have now ensured that all figures are properly displayed.

In panel E, the authors are making two clusters with PCA projections on single waveforms. Does this mean that the PCA is only applied to the main waveforms, i.e. the ones obtained where the amplitudes are peaking the most? This is not really clear from the methods, but if this is the case, then this approach is a bit simplistic and does not really match state-of-the-art solutions. Spike waveforms are quite often, especially with such high-density arrays, covering multiple channels at once, and thus the extracellular patterns triggered by the single units on the MEA are spatio-temporal motifs occurring on several channels. This is why, in modern spike sorters, the information in a local neighbourhood is often kept to be projected, via PCA, on the lower-dimensional space before clustering. Information on a single channel only might not be informative enough to disambiguate sources. Can the authors comment on that, and what is the exact spatial resolution of the 3Brain device? The way the authors are performing the SVD should be clarified in the methods section. Is it on a single channel, and/or on multiple channels in a local neighbourhood?

We agree with the reviewer that it would be useful to have the option of performing PCA on several channels at once, since spikes can occur at several channels at the same time. We have now added a routine to SpikeMAP that allows users to define a radius around individual channels prior to performing PCA. The text was clarified as follows (line 131):

“The SpikeMAP suite also offers a routine to select a radius around individual channels in order to enter groups of adjacent channels in PCA.”

(5) About the isolation of the single units, here again, I think the manuscript lacks some technical details. The authors are saying that they are using a k-means cluster analysis with k=2. This means that the authors are explicitly looking for 2 clusters per electrode? If so, this is a really strong assumption that should not be held in the context of spike sorting, because, since it is a blind source separation technique, one can not pre-determine in advance how many sources are present in the vicinity of a given electrode. While the illustration in Figure 2E is ok, there is no guarantee that one can not find more clusters, so why this choice of k=2? Again, this is why most modern spike sorting pipelines do not rely on k-means, to avoid any hard-coded number of clusters. Can the authors comment on that?

We clarified the text as follows (line 135):

“In SpikeMAP, the optimal number of k-means clusters can be chosen by a Calinski-Harabasz criterion (Calinski and Harabasz, 1974) or pre-selected by the user.”

(6) I'm surprised by the linear decay of the maximal amplitude as a function of the distance from the soma, as shown in Figure 2H. Is it really what should be expected? Based on the properties of the extracellular media, shouldn't we expect a power law for the decay of the amplitude? This is strange that up to 100um away from the soma, the max amplitude only dropped from 260 to 240 uV. Can the authors comment on that? It would be interesting to plot that for all neurons recorded, in a normed manner V/max(V) as function of distances, to see what the curve looks like.

We added Supplemental Figure 1 showing the drop in voltage over all putative somas (N=1,950) of one recording, after excluding somas with an increase voltage away from electrode peak and computing normed values V/max(V). We see a distribution of slopes as well as intercepts across somas, showing some variability across recordings sites. As the reviewer suggests, it is possible that a power-law describes these data better than a linear function, and this would need to be investigated further by quantitatively comparing the fit of these functions.

(7) In Figure 3A, it seems that the total number of cells is rather low for such a large number of electrodes. What are the quality criteria that are used to keep these cells? Did the authors exclude some cells from the analysis, and if yes, what are the quality criteria that are used to keep cells? If no criteria are used (because none are mentioned in the Methods), then how come so few cells are detected, and can the authors convince us that these neurons are indeed "clean" units (RPVs, SNRs, ...)?

The reviewer is correct to point out that a number of stringent criteria were employed to exclude some putative cells. We now outline these criteria directly in the text (line 161):

“ At different steps in the process, conditions for rejecting spikes can be tailored by applying: (1) a stringent threshold to filtered voltages; (2) a minimal cut-off on the signal-to-noise ratio of voltages (see Supplemental Figure 2); (3) an LDA for cluster separability; (4) a minimal spike rate to putative neurons; (5) a Hartigan statistical dip test to detect spike bursting; (6) a decrease in voltage away from putative somas; and (7) a maximum spike-count correlation for nearby channels. Together, these criteria allow SpikeMAP users the ability to precisely control parameters relevant to automated spike sorting.”

Further, we provide SNRs of individual channels (Supplemental Figure 2), and added to the SpikeMAP software the ability to apply a minimal criterion based on SNR.

(8) Still in Figure 3A, it looks like there is a bias to find inhibitory cells at the borders, since they do not appear to be uniformly distributed over the MEA. Can the authors comment on that? What would be the explanation for such a behaviour? It would be interesting to see some macroscopic quantities on Excitatory/Inhibitory cells, such as mean firing rates, averaged SNRs... Because again, in Figure 3C, it is not clear to me that the firing rates of inhibitory cells are higher than Excitatory ones, whilst they should be in theory.

We have added figures showing the distribution of E and I firing rates across a population of N=1,950 putative cells (Supplemental Figure 3). Firing rates of inhibitory neurons are marginally higher than excitatory neurons, and both E and I follow an approximately exponential distribution of rates.

Reviewer may be right that there are more I neurons at borders in Fig.3B because injections were done in medial prefrontal cortex, so this may reflect an experimental artefact related to a high probability of activating I neurons in locations where the opsin was activated. We added a sentence to the text to clarify this point (line 201):

“It is possible that the spatial location of putative I cells reflects the site of injection of the opsin in medial prefrontal cortex.”

(9) For Figure 3 in general, I would have performed an exhaustive comparison of putative cells found by spikeMAP and other sorters. More precisely, I think that to prove the point that spikeMAP is indeed bringing something new to the field of spike sorting, the authors should have compared the performances of various spike sorters to discriminate Exc vs Inh cells based on their ground truth recordings. For example, either using Kilosort [Pachitariu et al, 2024, 10.1038/s41592-024-02232-7], or some other sorters that might be working with such large high-density data [Yger et al, 2018, 10.7554/eLife.34518].

The reviewer is correct to point out that our the spike-sorting portion of our pipeline shares similarities with related approaches. Other aspects, however, are unique to SpikeMAP. We have clarified the text accordingly:

“In sum, SpikeMAP provides an end-to-end pipeline to perform spike-sorting on high-density multielectrode arrays. Some elements of this pipeline are similar to related approaches (Table 1), including the use of voltage filtering, PCA, and k-means clustering. Other elements are novel, including the use of spline interpolation, LDA, and the ability to identify putative excitatory and inhibitory cells.”

(10) Figure 4 has a big issue, and I guess the panels A and B should be redrawn. I don't understand what the red rectangle is displaying.

Again, we apologize for the rendering issues in the Figures that occurred during conversion into PDF format. We have now ensured that all figures are properly displayed.

(11) I understand that Figure 4 is only one example, but I have a hard time understanding from the manuscript how many slices/mices were used to obtain the GT data? I guess the manuscript could be enhanced by turning the data into an open-access dataset, but then some clarification is needed. How many flashes/animals/slices are we talking about? Maybe this should be illustrated in Figure 4, if this figure is devoted to the introduction of the GT data.

Details of the open access data are now provided in Supplemental Table 1. We also clarified Figure 5B:

“Quantification of change in firing rate following optogenetic stimulation. Average firing rates are taken over four recordings obtained from 3 mice.”

(12) While there is no doubt that GT data as the ones recorded here by the authors are the most interesting data from a validation point of view, the pretty low yield of such experiments should not discourage the use of artificially generated recordings such as the ones made in [Buccino et al, 2020, 10.1007/s12021-020-09467-7] or even recently in [Laquitaine et al, 2024, 10.1101/2024.12.04.626805v1]. In these papers, the authors have putative waveforms/firing rate patterns for excitatory and inhibitory cells, and thus, the authors could test how good they are in discriminating the two subtypes.

We agree with the reviewer that it would be worthwhile for future work to apply SpikeMAP to artificially generated spike trains, and have added the following (line 328):

“Another avenue could involve applying SpikeMAP on artificially generated spike data (Buccino & Einevoll 2021; Laquitaine et al., 2024).”

**Reviewer #1 (Recommendations for the authors):**
(1) Line 154 seems to include a parenthetical expression left over from editing: "sensitive to noise (contamination? Better than noise?) generated by the signal of proximal units." See also line 186: "use (reliance?) of light-sensitive" and line 245: "In the absence of synaptic blockers (right?)," and line 270: "the size of the data prevents manual intervention (curation?)." Check carefully for all parentheses like that, which should be removed.

Thank you for pointing this out. We have revised the text and removed parenthetical expressions left over from editing.

(2) In lines 285-286, you state that: "k-mean clustering of spike waveform properties best differentiated the two principal classes of cells..." But I could not find where you compared k-means clustering to other methods. I think you just argued that k-means seemed to work well, but not better than, another method. If that is so, then you should probably rephrase those lines.

The reviewer is correct that direct comparisons are not performed here, hence we removed this sentence.

(3) Methods section, E/I classification, lines 396-405: You give us figures on what fraction was E and I (PV subtype) (94.75% and 5.25%), but there is more that you could have said. First of all, what is the expected fraction of parvalbumin-sensitive interneurons in the cortex - is it near 5%?

We clarified the text as follows (line 444): “This number is close to the expected percentage of PV interneurons in cortex (4-6%) (Markram et al. 2004).”

Second, how would these percentages change if you altered the threshold from 3 s.d. to something lower, like 2 s.d.? Giving us some idea of how the threshold affects the fraction of PV interneurons could give us an idea of whether this method agrees with our expectations or not.

While SpikeMAP offers the flexibility to set the voltage threshold manually, we opted for a stringent threshold to demonstrate the capabilities of the software. As seen in Figure 2D, at 2 and 3 s.d., the signal is largely accounted for by Gaussian noise, while deviation from noise arises around 4 s.d. We clarified the text as follows (line 120):

“At a threshold of -3 , the signal could be largely accounted for by Gaussian noise, while a separation between signal and noise began around a threshold of -4 ”

Third, did the inhibitory neurons identified by this optogenetic method also have narrow spike widths at half amplitude? Could you do a scatterplot of all the spike widths and inter-peak distances that had color-coded dots for E and I based on your optogenetic method?

We have added a scatterplot (Supplemental Figure 5).

(4) Can you compare your methods with others now widely in use, like, for example, Spiking Circus or Kilosort? You do that in Table 1 in terms of features, but not in terms of performance. For example, you could have applied Kilosort4 to your data from the 4096 electrode array and seen how often it sorted the same neurons that SpikeMAP did. I realize this could not give you a comparison of how many were E/I, but it could tell you how close your numbers of neurons agreed with their numbers. Were your numbers within 5% of each other? This would be helpful for groups who are already using Kilosort4.

As mentioned ealier, packages listed in Table 1 do not provide an identification of putative E/I neurons on high-density electrode arrays. To facilitation the integration of SpikeMAP with other spike sorting packages, our suite now provides a stand-alone module to perform E/I identification. This is now mentioned in the text (see earlier comment).

**Reviewer #2 (Recommendations for the authors):**
I would encourage the authors to decide what the paper is about: is it about a new sorting method (and if yes, more tests/benchmarks are needed to explain the pros and the cons of the pipelines, and the Methods need to be expanded). Or is it about the new data for Ground Truth validation, and again, if yes, then maybe explain more what they are, how many slices/mice/cells, ... Maybe also consider making the data available online as an open dataset.

We agree with the reviewer that the paper is best slated toward ground truth validation of E/I identification. We now specify how many slices/mice/cells etc. (see Supplemental Table 1) and make the data available online as open source.